# Opposing roles for Bmp signalling during the development of electrosensory lateral line organs

Alexander S Campbell[1], Martin Minařík[1], Roman Franěk[2], Michaela Vazačová[2], Miloš Havelka[2], David Gela[2], Martin Pšenička[2], Clare VH Baker[1]*

[1]Department of Physiology, Development & Neuroscience, University of Cambridge, Cambridge, United Kingdom; [2]Faculty of Fisheries and Protection of Waters, Research Institute of Fish Culture and Hydrobiology, University of South Bohemia in České Budějovice, Vodňany, Czech Republic

**Abstract** The lateral line system enables fishes and aquatic-stage amphibians to detect local water movement via mechanosensory hair cells in neuromasts, and many species to detect weak electric fields via electroreceptors (modified hair cells) in ampullary organs. Both neuromasts and ampullary organs develop from lateral line placodes, but the molecular mechanisms underpinning ampullary organ formation are understudied relative to neuromasts. This is because the ancestral lineages of zebrafish (teleosts) and *Xenopus* (frogs) independently lost electroreception. We identified *Bmp5* as a promising candidate via differential RNA-seq in an electroreceptive ray-finned fish, the Mississippi paddlefish (*Polyodon spathula*; Modrell et al., 2017, *eLife* 6: e24197). In an experimentally tractable relative, the sterlet sturgeon (*Acipenser ruthenus*), we found that *Bmp5* and four other Bmp pathway genes are expressed in the developing lateral line, and that Bmp signalling is active. Furthermore, CRISPR/Cas9-mediated mutagenesis targeting *Bmp5* in G0-injected sterlet embryos resulted in fewer ampullary organs. Conversely, when Bmp signalling was inhibited by DMH1 treatment shortly before the formation of ampullary organ primordia, supernumerary ampullary organs developed. These data suggest that Bmp5 promotes ampullary organ development, whereas Bmp signalling via another ligand(s) prevents their overproduction. Taken together, this demonstrates opposing roles for Bmp signalling during ampullary organ formation.

*For correspondence:
cvhb1@cam.ac.uk

**Competing interest:** The authors declare that no competing interests exist.

## Editor's evaluation

This fundamental study provides new insight into the molecular signalling pathways that govern the formation of electrosensory ampullary organs in a non-model organism, the sterlet sturgeon. By using a combination of targeted gene knock-out and chemical inhibition, the study provides overall convincing evidence for differential roles of BMP signaling in lateral-line development. The study is particularly helpful for understanding the development of a still enigmatic sensory system, and for its evolutionary implications.

## Introduction

The lateral line system is an evolutionarily ancient sensory system found in fishes and aquatic-stage amphibians (**Bullock et al., 1983**; **Northcutt, 1997**; **Mogdans, 2021**). There are two distinct types of lateral line organs in the skin. Neuromasts, arranged in characteristic lines across the head and trunk, detect local water movement via mechanosensory hair cells ('touch at a distance') whose apical surfaces are embedded in a gelatinous cupula (**Cernuda-Cernuda and García-Fernández, 1996**; **Montgomery**

*et al., 2014*; *Pickett and Raible, 2019*; *Mogdans, 2021*). In non-teleost electroreceptive fishes and amphibians, fields of electrosensory ampullary organs flank some or all of the neuromast lines on the head (*Bullock et al., 1983*; *Baker et al., 2013*; *Crampton, 2019*). The electrosensory division of the lateral line system was independently lost in several lineages, for example, those leading to frogs/toads and to teleost fishes (although electroreception with physiologically distinct electroreceptors independently evolved multiple times in a few groups of teleost fishes) (*Bullock et al., 1983*; *Baker et al., 2013*; *Crampton, 2019*). Although salamanders (for example, the axolotl) are electroreceptive, the primary anamniote lab models, *Xenopus* and zebrafish, as well as other lab model teleosts such as medaka and cavefish, only have the mechanosensory division.

Non-teleost ampullary organs have a 'flask-shaped' chamber with a sensory epithelium at the base, connected to a pore in the epidermis via a canal filled with an electrically conductive jelly (*Jørgensen, 2011*; *Josberger et al., 2016*; *Zhang et al., 2018*). Ampullary electroreceptor cells are modified hair cells (*Jørgensen, 2005*; *Baker and Modrell, 2018*; *Baker, 2019*) that respond to weak cathodal (exterior-negative) electric fields, primarily for detecting prey or avoiding predators (*Bodznick and Montgomery, 2005*; *Crampton, 2019*; *Leitch and Julius, 2019*; *Chagnaud et al., 2021*). Both neuromasts and ampullary organs contain several types of support cells that flank the sensory receptor cells: these have a range of support functions including secretion (see, for example, *Cernuda-Cernuda and García-Fernández, 1996*; *Camacho et al., 2007*; *Russell et al., 2022*). In zebrafish neuromasts, both active and quiescent stem cell populations have been identified amongst the various support cell populations, which differentiate into hair cells during homeostasis and after injury (see *Lush and Piotrowski, 2014*; *Lush et al., 2019*; *Undurraga et al., 2019*).

Neuromasts, ampullary organs and their afferent neurons all develop from a series of lateral line placodes (thickened patches of neurogenic ectoderm) on the head (*Northcutt, 1997*; *Piotrowski and Baker, 2014*; *Baker, 2019*). There are six bilateral pairs of lateral line placodes: the anterodorsal, anteroventral and otic lateral line placodes form rostral to the otic vesicle, whereas the middle, supratemporal and posterior lateral line placodes form caudal to the otic vesicle (*Northcutt, 1997*; *Piotrowski and Baker, 2014*; *Baker, 2019*). Neuroblasts delaminate from the pole of each placode lying closest to the otic vesicle; they form afferent bipolar neurons (which coalesce in lateral line ganglia) whose peripheral axons accompany the placode as it continues to develop and form sensory organs (see *Piotrowski and Baker, 2014*; *McGraw et al., 2017*; *Chitnis, 2021*).

Neuromasts on the trunk originate from the posterior lateral line placode, whose development has been most intensively studied in the teleost zebrafish (see, for example, *Piotrowski and Baker, 2014*; *McGraw et al., 2017*; *Chitnis, 2021*). Initially, it gives rise to an early-migrating primordium (primI) that migrates as a cell-collective along the trunk, depositing neuromasts and a line of interneuromast cells that act as progenitors for later-forming neuromasts (reviewed by *Piotrowski and Baker, 2014*). A day later, another placode develops in the same position, which gives rise to two primordia: primD migrates dorsally to give rise to a dorsal line of neuromasts; the other (primII) migrates along the same pathway as primI, depositing secondary neuromasts in between the primary neuromasts left behind by primI (reviewed by *Piotrowski and Baker, 2014*). The migrating posterior lateral line primordium is closely followed by the growth cones of afferent lateral line axons and their associated Schwann cells (*Metcalfe, 1985*; *Gilmour et al., 2002*; *Gilmour et al., 2004*).

In non-teleosts, the other lateral line placodes do not migrate, but rather elongate over the head to form sensory ridges that eventually fragment, leaving a line of neuromasts along the centre of the ridge (see *Winklbauer, 1989*; *Piotrowski and Baker, 2014*). In electroreceptive species, ampullary organs form later than neuromasts, in fields flanking the line of neuromasts (*Northcutt, 2005*; *Baker et al., 2013*; *Piotrowski and Baker, 2014*). Just as in the migrating posterior lateral line primordium, afferent axons and associated Schwann cells accompany the elongating primordia (*Winklbauer, 1989*; *Northcutt, 2005*; *Piotrowski and Baker, 2014*).

Given the loss of electroreception in the lineages leading to frogs/toads and teleosts, we used a chondrostean ray-finned fish, the Mississippi paddlefish (*Polyodon spathula*), which has more ampullary organs than any other species (*Chagnaud et al., 2021*), as a model to study ampullary organ development (*Modrell et al., 2011a*; *Modrell et al., 2011b*; *Modrell et al., 2017a*; *Modrell et al., 2017b*; *Minařík et al., 2024a*). To identify candidate genes potentially involved in ampullary organ development, we performed a differential bulk RNA-seq screen at late-larval stages, comparing gene expression in fin tissue (which lacks lateral line organs) versus operculum tissue (which has many

ampullary organs and some neuromasts). This resulted in a lateral line-enriched gene-set containing almost 500 candidate genes enriched by at least twofold in paddlefish opercular versus fin tissue (*Modrell et al., 2017a*). Expression analysis of a range of candidate genes from this dataset and other candidates important for hair cell development (*Modrell et al., 2017a*; *Modrell et al., 2017b*; *Minařík et al., 2024a*), together with small-molecule manipulation of the Fgf and Notch signalling pathways (*Modrell et al., 2017b*), suggested that electoreceptors are closely related to hair cells and that the mechanisms underlying their development are highly conserved. To enable further investigation of gene function in ampullary organ and electroreceptor development, we moved to a more experimentally tractable chondrostean with a much longer spawning season: a small sturgeon, the sterlet (*Acipenser ruthenus*). Investigation of additional candidate genes from the paddlefish lateral line-enriched dataset in paddlefish and sterlet identified both mechanosensory-restricted and electrosensory-restricted transcription factor gene expression (*Modrell et al., 2017a*; *Minařík et al., 2024a*). We recently used CRISPR/Cas9-mediated mutagenesis in G0-injected sterlet embryos to identify a conserved requirement for the 'hair cell' transcription factor Atoh1 in electroceptor formation and identified a role for mechanosensory-restricted Foxg1 in blocking ampullary organ formation within neuromast lines (preprint, *Minařík et al., 2024b*).

One gene present in the paddlefish lateral line-enriched gene set was the Bmp ligand gene *Bmp5* (2.5-fold enriched in late-larval paddlefish operculum versus fin tissue; *Modrell et al., 2017a*). Here, we aimed to investigate the expression and function of *Bmp5* and Bmp signalling in the formation of sterlet lateral line organs. This led to our uncovering opposing roles for Bmp signalling during ampullary organ formation.

## Results

### *Bmp5* is expressed early in developing ampullary organs and later in neuromasts

The only Bmp ligand gene in the paddlefish lateral line organ-enriched gene-set was *Bmp5* (2.5-fold enriched in late-larval paddlefish operculum versus fin tissue; *Modrell et al., 2017a*). Wholemount in situ hybridisation (ISH) in sterlet yolk-sac larvae from stage 37 (hatching occurs at stage 36) to the onset of independent feeding at stage 45 (staging according to *Dettlaff et al., 1993*), revealed the time-course of *Bmp5* expression relative to the maturation of neuromasts and ampullary organs. The latter was shown by ISH for *Cacna1d*, encoding a voltage-gated calcium channel (Ca$_v$1.3) expressed by differentiated hair cells and electroreceptors (and taste-buds, for example on the barbels) (*Modrell et al., 2017a*; *Minařík et al., 2024a*). Within each lateral line primordium, neuromasts form before ampullary organs and hair cells differentiate much earlier than electroreceptors (*Minařík et al., 2024a*).

At stage 37, faint *Bmp5* expression was seen within developing gill filaments but there was no detectable lateral line expression (*Figure 1A and B*), despite the presence of some differentiated neuromasts (i.e., with *Cacna1d*-expressing hair cells; *Figure 1C and D*). By stage 40, strong *Bmp5* expression was visible in mature neuromasts as well as ampullary organ primordia (*Figure 1E and F*; compare with *Cacna1d* expression in *Figure 1G and H*, which shows that few electroreceptors have differentiated at this stage). At stage 42, *Bmp5* expression was seen in mature ampullary organs but seemed weaker in neuromasts (*Figure 1I and J*; compare with *Cacna1d* expression in *Figure 1K and L*). By stage 45, *Bmp5* expression persisted in ampullary organs, although this seemed weaker than at stage 40, and was no longer seen in most neuromasts on the head (*Figure 1M and N*; compare with *Cacna1d* expression in *Figure 1O and P*). On the trunk, *Bmp5* expression was seen in a subset of regularly spaced neuromasts in the main body line, as well as the dorsal line deposited by primD, with stronger expression in more rostral (i.e., earlier-deposited) neuromasts (*Figure 1Q*; compare with *Cacna1d* expression in all trunk neuromasts in *Figure 1R*). The *Bmp5*-expressing neuromasts in the main body line are secondary neuromasts deposited by the later-migrating primII, which are offset slightly dorsally to those deposited earlier by primI, which are *Bmp5*-negative (compare with *Cacna1d* expression in *Figure 1R*; examples of primII-deposited neuromasts are highlighted). *Figure 1S-V* show schematic summaries of cranial neuromast and ampullary organ development at similar stages (from *Minařík et al., 2024a*).

Overall, these results suggest that *Bmp5* is expressed early within ampullary organ primordia and maintained in mature ampullary organs at least through to the onset of independent feeding at stage

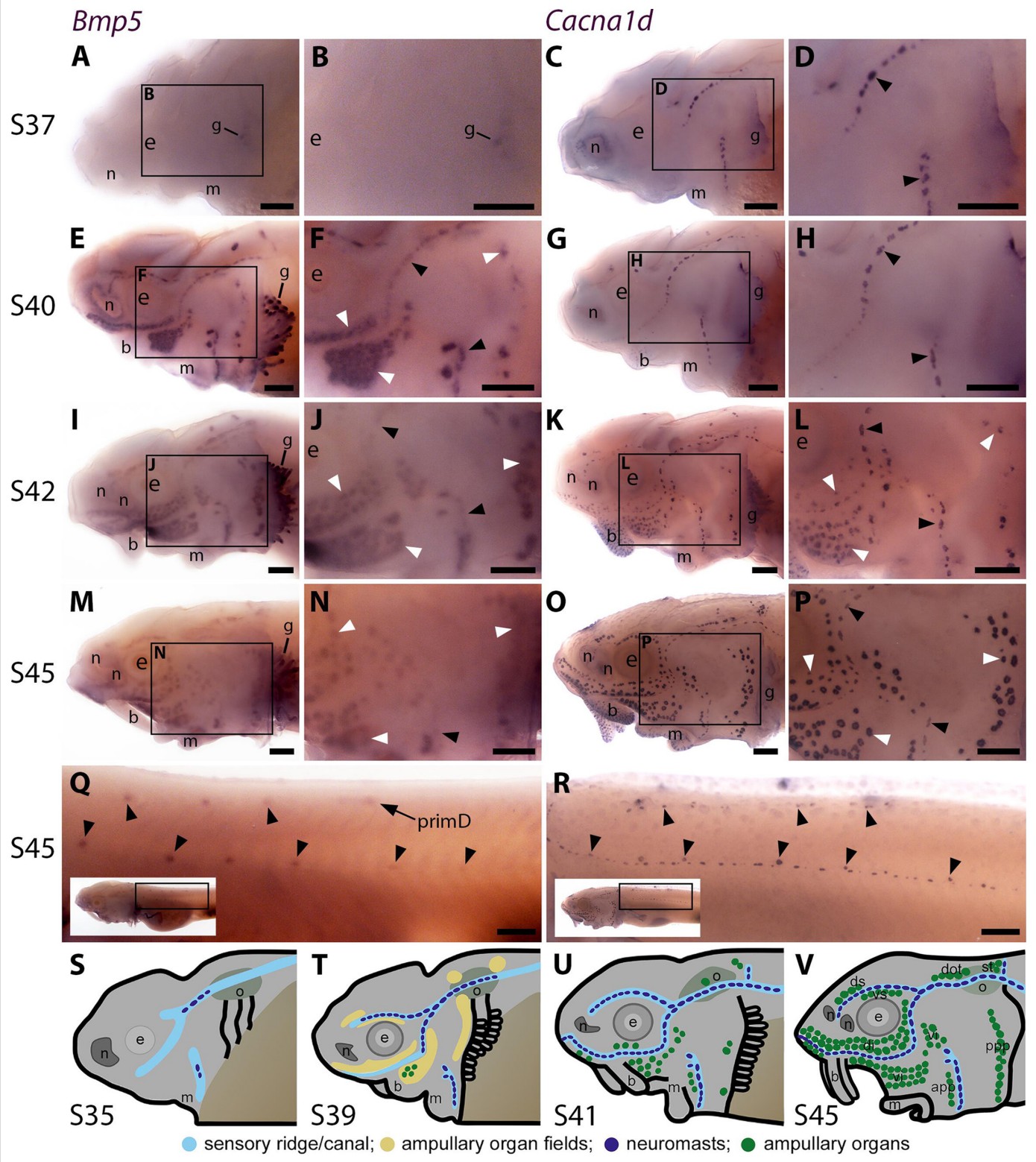

**Figure 1.** Sterlet *Bmp5* is expressed early in developing ampullary organs and transiently in mature neuromasts. (**A–R**) In situ hybridisation in sterlet for *Bmp5* or the hair cell and electroreceptor marker *Cacna1d*, which labels mature neuromasts and ampullary organs (also expressed in taste buds on the barbels). Black arrowheads indicate examples of developing neuromasts; white arrowheads indicate examples of developing ampullary organs. (**A–D**) At stage 37, *Bmp5* expression is only detectable in developing gill filaments (**A,B**) although *Cacna1d*-positive neuromasts are present (**C,D**). (**E–H**) At stage 40, *Bmp5* is expressed in neuromasts and ampullary organ primordia (**E,F**); only a few *Cacna1d*-positive ampullary organs are present at this

*Figure 1 continued on next page*

*Figure 1 continued*

stage (**G, H**). (**I–L**) At stage 42, *Bmp5* is expressed in mature ampullary organs and more weakly in neuromasts (**I,J**); compare with *Cacna1d* expression (**K,L**). (**M–P**) At stage 45 (onset of independent feeding), *Bmp5* expression is weaker in ampullary organs and no longer detectable in most neuromasts (**M,N**); compare with *Cacna1d* expression (**O,P**). (**Q,R**) At stage 45 on the trunk, *Bmp5* expression is visible in primII-deposited secondary neuromasts (more strongly in more rostral neuromasts) as well as in primD and neuromasts of the dorsal line (**Q**). Compare with *Cacna1d* expression in all neuromasts (**R**): arrowheads indicate examples of dorsal-line neuromasts and primII-deposited secondary neuromasts (offset a little dorsal to the line of primI-deposited primary neuromasts). Low-power insets show the location of these trunk regions. (**S–V**) Schematic depictions of sterlet lateral line organ development at similar stages (stages 35, 39, 41, 45), previously published in *Minařík et al., 2024a*. Abbreviations: app, anterior preopercular ampullary organ field; b, barbel; di, dorsal infraorbital ampullary organ field; dot, dorsal otic ampullary organ field; ds, dorsal supraorbital ampullary organ field; e, eye; gf, gill filaments; m, mouth; n, naris; o, otic vesicle; ppp, posterior preopercular ampullary organ field; prim, migrating lateral line primordium (primI, primary; primII, secondary; primD, dorsal); S, stage; st, supratemporal ampullary organ field; vi, ventral infraorbital ampullary organ field; vs, ventral supraorbital ampullary organ field. Scale bar: 250 µm.

45. In contrast, *Bmp5* only seems to be expressed in mature neuromasts, after the onset of hair cell differentiation, and then only transiently.

## The Bmp signalling pathway is active throughout the developing sterlet lateral line system

To investigate where and when the Bmp signalling pathway is active during sterlet lateral line organ development, we performed wholemount immunohistochemistry using an antibody raised against human phospho-SMAD1/5/9 (pSMAD1/5/9; SMAD9 was formerly known as SMAD8), as a proxy for Bmp signalling (*Schmierer and Hill, 2007*).

At stage 30 (the earliest stage examined), faint pSMAD1/5/9 immunoreactivity was seen in the region of the anteroventral lateral line primordium (*Figure 2A and B*; compare with Sox2 immunoreactivity at stage 32 in *Figure 2—figure supplement 1A and B*; Sox2 is expressed in lateral line primordia and maintained in supporting cells; *Hernández et al., 2007*; *Modrell et al., 2017a*; *Minařík et al., 2024a*). By stage 34, pSMAD1/5/9 immunoreactivity was detectable in lateral line primordia, with a ring pattern around developing neuromast primordia in the otic/anterodorsal and anteroventral primordia (*Figure 2C and D*; compare with stage 35 Sox2 expression in *Figure 2—figure supplement 1C and D*). (The first *Cacna1d*-positive differentiated hair cells are seen in this region at stage 35; *Minařík et al., 2024a*.) At stage 36 (*Figure 2E and F*), weak pSMAD1/5/9 immunoreactivity was still seen around developing neuromasts (compare with stage 37 *Cacna1d* expression in *Figure 1C and D*; stage 37 Sox2 immunoreactivity in *Figure 2—figure supplement 1E and F*), but we were intrigued to see more prominent immunoreactivity in a filamentous pattern that seemed most likely to correspond to lateral line nerves (*Figure 2E and F*). This pattern continued at stages 38 and 40 (*Figure 2G–J*); indeed pSMAD1/5/9 immunoreactive collaterals seemed to be developing from the infraorbital nerve between stages 38 and 40 (*Figure 2G–J*). At stage 40, diffuse immunoreactivity was also seen in regions flanking the nerves where ampullary organ primordia are forming (*Figure 2I and J*; compare with stage 40 *Bmp5* expression in *Figure 1E and F*). (Strong pSMAD1/5/9 immunoreactivity was also seen in the barbel primordia, around the nares and mouth, in gill filaments, and between stages 36 and 40 in a patch between the barbels and the otic vesicle that likely represents a muscle, the *m. protractor hyomandibulae*; *Warth et al., 2018*.)

By stage 42, pSMAD1/5/9 immunoreactivity was visible in ampullary organs and much less prominent in lateral line nerves (*Figure 2K and L*; compare with stage 42 Sox2 expression in *Figure 2—figure supplement 1I and J*). At stages 43 and 45, pSMAD1/5/9 immunoreactivity was more clearly visible in neuromasts as well as ampullary organs, in all cases at the periphery rather than centre of each organ (*Figure 2M–P*; compare with the hair cell/electroreceptor marker *Cacna1d* at stage 42 in *Figure 1K and L* and at stage 45 in *Figure 1O and P*, and the supporting cell marker Sox2 at stages 42 and 45 in *Figure 2—figure supplement 1I–L*). This peripheral pattern of pSMAD1/5/9 immunoreactivity suggests that Bmp signalling is active in supporting cells rather than receptor cells. Also at stages 43 and 45, we noted that pSMAD1/5/9 immunoreactivity on the head seemed to be particularly strong in the supraorbital and infraorbital neuromast lines (compare with Sox2 expression at stages 42 and 45 in *Figure 2—figure supplement 1I–L*).

pSMAD1/5/9 immunoreactivity was also prominent in the migrating lateral line primordia on the trunk. At stage 39, pSMAD1/5/9 immunoreactivity was seen in primI and a diffuse but somewhat

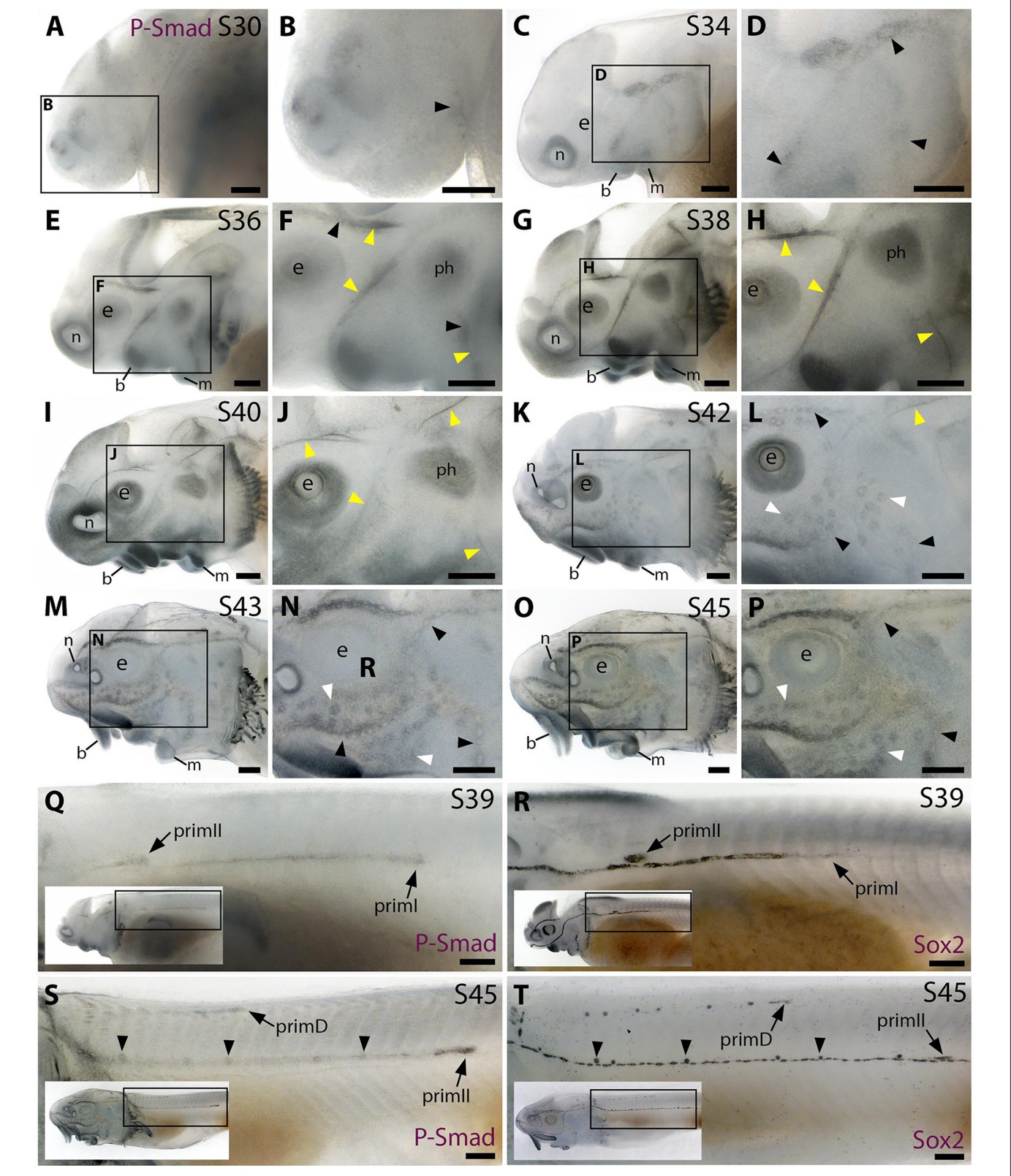

**Figure 2.** The Bmp signalling pathway is active throughout the developing lateral line system in sterlet. Immunostaining in sterlet. Black arrowheads indicate examples of developing neuromasts; white arrowheads indicate examples of developing ampullary organs; yellow arrowheads indicate lateral line nerves. (**A–P**) Immunoreactivity on the head for phospho-SMAD1/5/9 (P-Smad) as a proxy for Bmp signalling activity. At stage 30 (**A,B**), weak immunoreactivity is seen in the region of the anteroventral lateral line primordium and by stage 34 (**C,D**) in lateral line primordia, with a ring pattern

*Figure 2 continued on next page*

*Figure 2 continued*

around developing neuromast primordia. At stages 36–40 (**E–J**), immunoreactivity is weak around developing neuromasts and prominent in lateral line nerves (yellow arrowheads). At stage 40 (**I,J**), diffuse immunoreactivity is also seen in regions flanking the nerves where ampullary organ primordia are forming. Non-lateral line immunoreactivity is present around the mouth and nares, in barbel primordia, gill filaments, and a patch that is most likely the developing muscle *m. protractor hyomandibulae*. Between stages 42 and 45 (**K–P**), immunoreactivity disappears in lateral line nerves and is increasingly detected at the periphery of ampullary organs and neuromasts (strongly in supraorbital and infraorbital neuromast lines). (**Q–T**) Immunostaining on the trunk (boxes on low-power insets indicate the location of the trunk regions shown). At stage 39 (**Q,R**), pSMAD1/5/9 immunoreactivity is seen in primI and a diffuse trail behind it, and in primII (**Q**). For comparison, Sox2 is expressed weakly in primI and strongly in primI-deposited neuromasts and interneuromast cells, plus primII (**R**). At stage 45 (**S,T**), pSMAD1/5/9 immunoreactivity is seen in primD and primII plus a weak trail behind it, with greater intensity at the periphery of primII-deposited neuromasts (**S**). For comparison, Sox2 expression is strong in primII, primD and all neuromasts; arrowheads indicate examples of primII-deposited neuromasts (**T**). Abbreviations: b, barbel; e, eye; f, fin; g, gill filaments; m, mouth; n, naris; ph, *m. protractor hyomandibulae*; prim, migrating lateral line primordium (primI, primary; primII, secondary; primD, dorsal); S, stage. Scale bar: 250 μm.

The online version of this article includes the following figure supplement(s) for figure 2:

**Figure supplement 1.** Sox2 expression shows the time-course of sterlet lateral line organ development.

continuous line trailing behind it, as well as in primII, which is located much further rostrally and a little dorsal to the main line (*Figure 2Q*; compare with Sox2 expression in *Figure 2R*). At stage 45, strong pSMAD1/5/9 immunoreactivity was seen in primII and primD, and more weakly along the path taken by primII, with increased intensity at the periphery of the primII-deposited neuromasts (*Figure 2S*; compare with Sox2 expression in all neuromasts in *Figure 2T* and with *Bmp5* expression in primII-deposited neuromasts in *Figure 1Q*). (For further comparison, *Figure 2—figure supplement 1M–P* show the positions of the different migrating primordia on the trunk via Sox2 expression at stages 38, 40, 42 and 45.)

Overall, these data suggest that Bmp signalling is active throughout lateral line development in the sterlet, including lateral line organ primordia and even lateral line nerves, and later at the periphery (rather than the centre) of maturing ampullary organs and neuromasts, suggesting activity in supporting cells rather than receptor cells.

## *Bmp4* is also expressed during sterlet lateral line organ development

*Bmp5* was the only gene encoding a Bmp ligand or receptor in the late-larval paddlefish lateral line-enriched gene-set (*Modrell et al., 2017a*). However, the timecourse and pattern of pSMAD1/5/9 immunoreactivity in the developing sterlet lateral line system was more extensive than *Bmp5* expression (compare *Figure 1* and *Figure 2*), suggesting other Bmp ligands must be expressed that were not enriched in the transcriptome of late-larval paddlefish operculum versus fin tissue (*Modrell et al., 2017a*). We therefore searched the pooled larval sterlet transcriptome that was available to us at the time, for additional Bmp pathway ligand and receptor genes for cloning and ISH. This enabled us to examine the expression of the ligand gene *Bmp4* and the type II receptor gene *Acvr2a*. Indeed, after these experiments were underway, a lateral line organ-enriched gene-set from stage 45 Siberian sturgeon (*Acipenser baerii*) was published that included *Bmp4* as well as *Bmp5* (*Wang et al., 2020*).

*Bmp4* expression was not evident in the developing sterlet lateral line at stage 37 or stage 38 (*Figure 3A–D*); the very faint expression in two widely spaced dorsal patches at stage 38 may be sensory patches in the otic vesicle (*Figure 3C and D*), or may represent early-forming neuromast primordia in the otic and supratemporal lines (see *Gibbs and Northcutt, 2004*). Stronger expression was also seen in fin, barbel and gill filament primordia, and around the nares (*Figure 3A–D*). By stage 40, diffuse *Bmp4* expression was observed in neuromast regions and fields of ampullary organ primordia on the head (*Figure 3E and F*; compare with *Bmp5* and *Cacna1d* expression in *Figure 1E–H*). Stronger expression was also seen on the trunk in primI, with much weaker expression in the most recently deposited neuromasts near to the primordium, plus a spot that most likely represents primII (*Figure 3G*; compare with stage 40 Sox2 expression on the trunk in *Figure 2—figure supplement 1N*). At stage 42, *Bmp4* expression in lateral line regions on the head was almost gone, with only faint expression remaining in the dorsal infraorbital ampullary organ field, just below the eye (*Figure 3H and I*). However, strong expression was seen in primII and primD on the trunk (*Figure 3J*; compare with stage 42 Sox2 expression in *Figure 2—figure supplement 1O*). By stage 45, *Bmp4* was no longer expressed in lateral line regions on the head, although expression remained in the barbels and gills (*Figure 3K–L*). As at stages 40 and 42, the migrating lateral line primordia on the trunk still

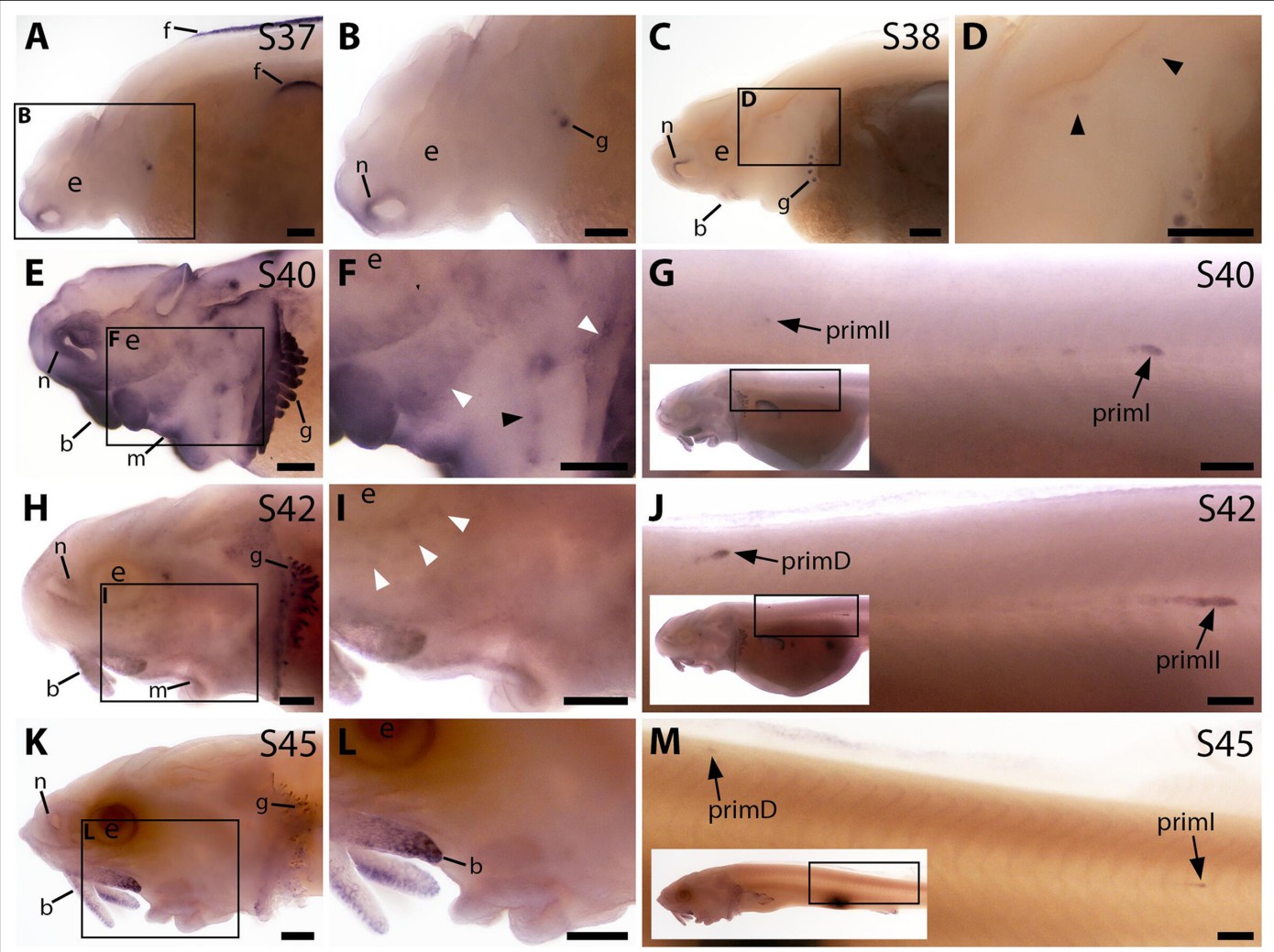

**Figure 3.** *Bmp4* is expressed transiently during sterlet lateral line organ development. In situ hybridisation in sterlet for *Bmp4*. Black arrowheads indicate examples of neuromast regions; white arrowheads indicate examples of ampullary organ regions. For images of the trunk, boxes on low-power insets delineate the location of the trunk regions shown. (**A,B**) At stage 37, *Bmp4* is not expressed in lateral line regions, although it is present around the nares and in fins and gill-filament primordia. (**C,D**) At stage 38, two dorsal spots of weak *Bmp4* expression may represent sensory patches in the otic vesicle or early-forming neuromast primordia in the otic and supratemporal lines. Expression is also present in the gills, nares and barbel primordia. (**E–G**) At stage 40, expression is seen on the head in neuromast regions and fields of ampullary organ primordia (**E,F**; compare with *Bmp5* and *Cacna1d* expression in *Figure 1E–H*). On the trunk, *Bmp4* is expressed in primI and the most recently deposited neuromasts behind it, and in primII (**G**). (**H–J**) At stage 42, *Bmp4* expression on the head has largely disappeared (**H,I**), apart from weak expression in the dorsal infraorbital field (arrowheads in **I**), although expression is still seen in gill filaments and barbels. On the trunk, expression is seen in primD and primII (**J**). (**K–M**) At stage 45, no lateral line expression is seen on the head (**K,L**), although weak expression persists in primD and primI on the trunk (**M**). Abbreviations: b, barbel; e, eye; f, fin; g, gill filaments; m, mouth; n, naris; prim, migrating lateral line primordium (primI, primary; primII, secondary; primD, dorsal); S, stage. Scale bar: 250 µm.

The online version of this article includes the following figure supplement(s) for figure 3:

**Figure supplement 1.** Sterlet *Acvr2a* is expressed in developing neuromasts and ampullary organs.

**Figure supplement 2.** Sterlet *Sostdc1* is expressed in neuromasts but only transiently in ampullary organs.

**Figure supplement 3.** Sterlet *Apcdd1* is expressed in ampullary organs and neuromasts during development.

expressed *Bmp4* at stage 45 (*Figure 3M*; compare with stage 45 *Sox2* expression in *Figure 2—figure supplement 1P*).

These data suggest that *Bmp4* plays a more transient role than *Bmp5* in lateral line organ development. Furthermore, most likely an as-yet unidentified Bmp ligand gene is expressed in lateral line primordia before either *Bmp5* or *Bmp4*, given that pSMAD1/5/9 immunoreactivity was detectable in lateral line primordia at stage 34 (*Figure 2C and D*).

The only Bmp receptor gene we examined was *Acvr2a*, encoding ActRIIA (activin A receptor type 2 A), a type II Bmp receptor that promiscuously binds multiple ligands including Bmp5 and Bmp4 (*Yadin et al., 2016*). *Acvr2a* was not expressed at stage 37 (*Figure 3—figure supplement 1A and B*), so other receptors must be involved in mediating Bmp signalling in lateral line primordia at this and earlier stages (see, for example, pSMAD1/5/9 immunoreactivity at stage 34 and stage 36; *Figure 2C–F*). By stage 38, although background levels were high, *Acvr2a* expression was detectable in developing neuromast regions (*Figure 3—figure supplement 1C and D*). By stage 40, *Acvr2a* was expressed at the periphery of ampullary organ primordia and neuromasts on the head (*Figure 3—figure supplement 1E and F*; compare with stage 40 *Bmp5* and *Cacna1d* expression in *Figure 1E–H*, and with stage 39 Sox2 expression in *Figure 2—figure supplement 1G and H*). Also at stage 40, *Acvr2a* was expressed in primI and a trailing line of cells behind it, plus a spot most likely representing primII (*Figure 3—figure supplement 1G*). This pattern persisted in both the head and trunk at stage 42, with expression now also seen primD and in the rostral trunk neuromasts deposited by primII (*Figure 3—figure supplement 1H–J*). By stage 45, *Acvr2a* expression appeared to be fading on the head, with only faint expression at the periphery of ampullary organs in a few areas (*Figure 3—figure supplement 1K and L*). However, expression continued in the trunk neuromast lines (*Figure 3—figure supplement 1M*). Overall, the *Acvr2a* expression pattern does not fully complement either *Bmp5* or *Bmp4* expression (compare with *Figure 1* and *Figure 3*, respectively), or pSMAD1/5/9 immunoreactivity (*Figure 2*). Hence, other type II receptor(s), as well of course as type I receptors, must be involved.

### *Sostdc1* and *Apcdd1*, encoding secreted dual Bmp/Wnt inhibitors, are expressed during sterlet lateral line organ development

Three genes encoding secreted Bmp inhibitors were present in the late-larval paddlefish lateral line-enriched gene-set (*Modrell et al., 2017a*): *Sostdc1*, *Apcdd1* and *Vwc2*. *Vwc2* was 4.5-fold lateral line-enriched (*Modrell et al., 2017a*), but ISH for this gene in sterlet was unsuccessful so it is not considered further.

Sostdc1 (sclerostin domain-containing 1; also known as Wise, Ectodin) is a secreted antagonist of both the Bmp and Wnt pathways (*Tong et al., 2022*). *Sostdc1* was 4.2-fold enriched in late-larval paddlefish operculum versus fin tissue (*Modrell et al., 2017a*). From stage 36 onwards, *Sostdc1* expression was seen in lines of differentiated neuromasts (and in gill filament primordia) on the head, and from stage 40 onwards, in the migrating primordia and neuromasts on the trunk (*Figure 3—figure supplement 2A–M*; compare with *Cacna1d* expression in *Figure 1* and Sox2 expression in *Figure 2—figure supplement 1E-P*). At stage 42, *Sostdc1* expression was also detected in ampullary organs (*Figure 3—figure supplement 2H and I*), but this had already disappeared by stage 45 (*Figure 3—figure supplement 2K and L*). These data suggest *Sostdc1* plays a persistent role within neuromasts, but any function in ampullary organ development is likely to be transient.

Apcdd1 (adenomatosis polyposis coli down-regulated 1) is also a secreted inhibitor of both the Bmp and Wnt pathways (*Vonica et al., 2020*). *Apcdd1* was 2.2-fold enriched in late-larval paddlefish operculum versus fin tissue (*Modrell et al., 2017a*). At stage 36, *Apcdd1* was not expressed in differentiated neuromast lines (*Figure 3—figure supplement 3A and B*), in contrast to *Sostdc1* at the same stage (*Figure 3—figure supplement 2A and B*). However, there was some *Apcdd1* expression in the region of the preopercular neuromast line, as well as outside the lateral line system: at the edge of the operculum, near the future barbel region and around the mouth (*Figure 3—figure supplement 3A and B*). At stage 38, more diffuse *Apcdd1* expression was seen in broader regions (*Figure 3—figure supplement 3C and D*). By stage 40, expression was visible around ampullary organ primordia and some neuromasts on the head (*Figure 3—figure supplement 3E and F*), as well as primI and primII on the trunk and a relatively short line of trailing cells behind primI (*Figure 3—figure supplement 3G*). By stage 42, *Apcdd1* expression on the head had largely resolved to the periphery of ampullary organs and neuromasts (*Figure 3—figure supplement 3H and I*; compare with stage 42 Sox2 expression in *Figure 2—figure supplement 1I and J*) and continued in the migrating primordia on the trunk and the short line of trailing cells behind primI (*Figure 3—figure supplement 3J*). At stage 45, this expression pattern largely persisted, although it seemed to be fading in the ventral infraorbital field (*Figure 3—figure supplement 3K and L*) and faint expression was also now seen along the main body line, potentially at the periphery of trunk

**Table 1.** sgRNAs used in this study.

The target sequences and sgRNA combinations used in this study are shown. The *Tyr* sgRNAs were previously published (preprint, *Minařík et al., 2024b*); the asterisk against *Tyr* sgRNAs 7 and 8 indicates that these sgRNAs were originally designed and published by *Stundl et al., 2022* as their *tyr* sgRNAs 3 and 4, respectively.

| Target Gene | sgRNA | Target Sequence | PAM | Combinations Used |
|---|---|---|---|---|
| *Bmp5* | 1 | TCACGCAGAAAAGCACAGGG | AGG | 1+2 + 3, 1+4 |
| | 2 | AGATGATGCCTGTTTGCCAG | GGG | 1+2 + 3, 2+3 |
| | 3 | GGCAAACGAGGAGGAAAACG | GGG | 1+2 + 3, 2+3 |
| | 4 | GTACAATGCCATGGCAAACG | AGG | 1+4 |
| *Tyr* | 1 | GGTGCCAAGGCAAAAACGCT | GGG | 1+2, 1+2 + 3+4 |
| | 2 | GATATCCCTCCATACATTAT | TGG | 1+2, 1+2 + 3+4 |
| | 3 | GATGTTTCTAAACATTGGGG | TGG | 1+2 + 3+4 |
| | 4 | GCTATGAATTTATTTTTTTC | AGG | 1+2 + 3+4 |
| | 5 | GCAAGGTATACGAAAGTTGA | CGG | 5+6 |
| | 6 | GATTGCAAGTTCGGCTTCTT | AGG | 5+6 |
| | 7* | GGTTAGAGACTTTATGTAAC | GGG | 7+8 |
| | 8* | GGCTCCATGTCTCAAGTCCA | AGG | 7+8 |

neuromasts (*Figure 3—figure supplement 3M*; compare with stage 45 Sox2 expression on the trunk in *Figure 2—figure supplement 1P*).

These data, especially the early, broad expression within ampullary organ fields and seemingly very late upregulation in neuromasts, suggest that Apcdd1 may be more important for ampullary organ development. In contrast, the pattern of *Sostdc1* expression (*Figure 3—figure supplement 2*) suggests its role may be more prominent during neuromast development. However, given the ability of Apcdd1 and Sostdc1 to inhibit both the Bmp and Wnt pathways (*Tong et al., 2022*; *Vonica et al., 2020*), we cannot be sure which of these pathway(s) either of these proteins may be antagonising during sterlet lateral line organ development.

## CRISPR/Cas9-mediated targeting of *Bmp5* results in fewer ampullary organs forming

Having established that the Bmp signalling pathway is active throughout lateral line organ development and that genes encoding two Bmp ligands, a type II receptor and two secreted dual Bmp/Wnt antagonists are expressed, we wanted to explore the role of Bmp signalling in lateral line development. *Bmp5* was chosen as a target for CRISPR/Cas9-mediated mutagenesis in G0-injected sterlet embryos owing to its earlier expression in ampullary organ primordia. We recently published our approach to CRISPR/Cas9 in sterlet (preprint, *Minařík et al., 2024b*). The experiments reported here were undertaken at the same time as those reported in *Minařík et al., 2024b*, preprint. Different 1–2 cell embryos from some of the same batches were injected with Cas9 protein complexed with different combinations of single-guide (sg) RNAs targeting *Bmp5*. Embryos targeted for the melanin-producing enzyme gene *tyrosinase* (*Tyr*) were used as negative controls: this yields a visible phenotype (i.e., defects in pigmentation), but should not affect other developmental processes (preprint, *Minařík et al., 2024b*).

Our *Bmp5* sgRNAs (*Table 1*; *Figure 4A*) were designed before the first chromosome-level sterlet genome was published (*Du et al., 2020*). Analysis of this genome showed that, rather than being functionally diploid as previously thought (from microsatellite data; *Ludwig et al., 2001*), the sterlet genome has retained a high level of tetraploidy, including around 70% of ohnologs (i.e., gene paralogs resulting from the independent whole-genome duplication in the sterlet lineage) (*Du et al., 2020*). Searching the reference genome (Vertebrate Genomes Project NCBI RefSeq assembly GCF_902713425.1) for *Bmp5* showed that both *Bmp5* ohnologs have been retained, on chromosomes 5 and 6, with 88.87% nucleotide identity in the coding sequence (and 95.60% amino acid identity). All of our *Bmp5* sgRNAs

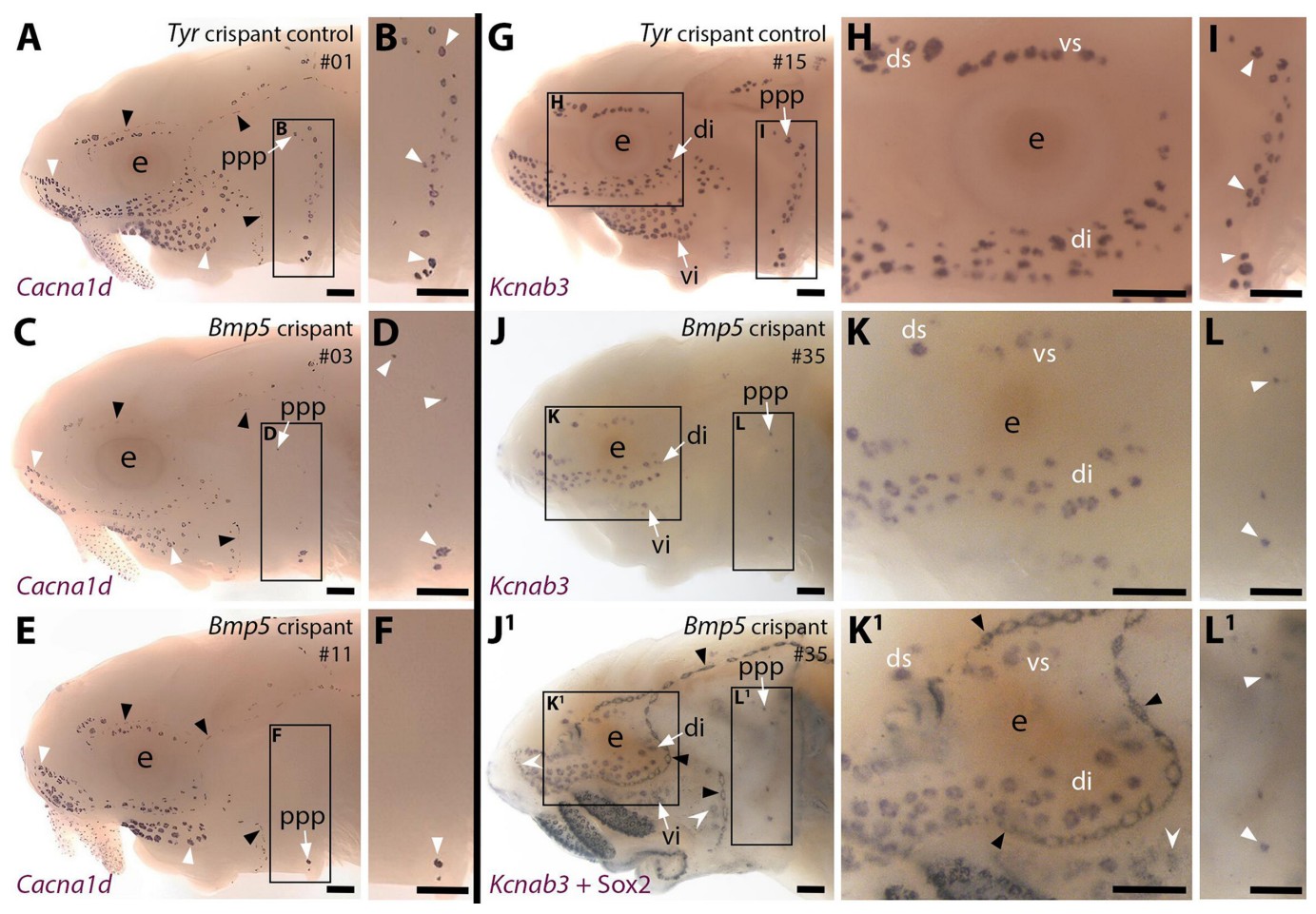

**Figure 4.** CRISPR/Cas9-mediated targeting of *Bmp5* leads to fewer ampullary organs developing. Sterlet crispants at stage 45 after in situ hybridisation (ISH) for the hair cell and electroreceptor marker *Cacna1d* (also expressed in taste buds on the barbels) or the electroreceptor-specific marker *Kcnab3*. All crispants shown are from the same batch of siblings/half-siblings (in vitro fertilisation used a mix of sperm from three different males). Black arrowheads indicate examples of neuromasts; white arrowheads indicate examples of ampullary organs. Crispants are numbered for cross-referencing with data provided for each crispant in ***Supplementary file 2***. (**A,B**) In a control *Tyr* crispant, *Cacna1d* expression shows the normal pattern of neuromast lines flanked by fields of ampullary organs. The higher power view shows the posterior preopercular ampullary organ field. (**C–F**) In *Bmp5* crispants, *Cacna1d* expression reveals fewer ampullary organs (compare **C,E** with **A**); this phenotype is particularly prominent in the posterior preopercular ampullary organ field (compare **D,F** with **B**). (**G–I**) In a control *Tyr* crispant, electroreceptor-specific *Kcnab3* expression shows the normal distribution of ampullary organs. (**J–L¹**) In a *Bmp5* crispant, *Kcnab3* expression shows fewer ampullary organs (compare **J-L** with **G-I**). Post-ISH Sox2 immunostaining for supporting cells (**J¹,K¹,L¹**) demonstrates that neuromasts have formed normally. Very few "additional" ampullary organs appeared (i.e., Sox2-positive, *Kcnab3*-negative ampullary organs: compare **J¹,K¹,L¹** with **I,J,K**); examples are indicated with indented white arrowheads. (Non-lateral line Sox2 expression is also seen in gill filaments and in taste buds on the barbels and around the mouth.) Abbreviations: di, dorsal infraorbital ampullary organ field; ds, dorsal supraorbital ampullary organ field; e, eye; ppp, posterior preopercular ampullary organ field; S, stage; vi, ventral infraorbital ampullary organ field; vs, ventral supraorbital ampullary organ field. Scale bar: 250 μm.

The online version of this article includes the following figure supplement(s) for figure 4:

**Figure supplement 1.** Examples of successful disruption of sterlet *Bmp5* by CRISPR/Cas9-mediated mutagenesis in G0-injected embryos.

fully match the ohnolog on chromosome 6. Relative to the ohnolog on chromosome 5, sgRNAs 2 and 4 (***Table 1***; ***Figure 4A***) each have a single-base mismatch, respectively, in positions 7 and 4 of the target sequence (PAM-distal), which should be tolerated (***Guo et al., 2014***; ***Rabinowitz and Offen, 2021***). However, our sgRNA 1 (***Table 1***; ***Figure 4A***) has two mismatched bases (at positions 3 and 12 of the target sequence) and sgRNA 3 (***Table 1***; ***Figure 4A***) has a single-base mismatch at position 20, adjacent to the PAM. Therefore, although we expect all our sgRNAs to target the chromosome 6 ohnolog, it is possible that only sgRNAs 2 and 4 successfully target the chromosome 5 ohnolog (***Guo et al., 2014***; ***Rabinowitz and Offen, 2021***). Nevertheless, given that all combinations of injected

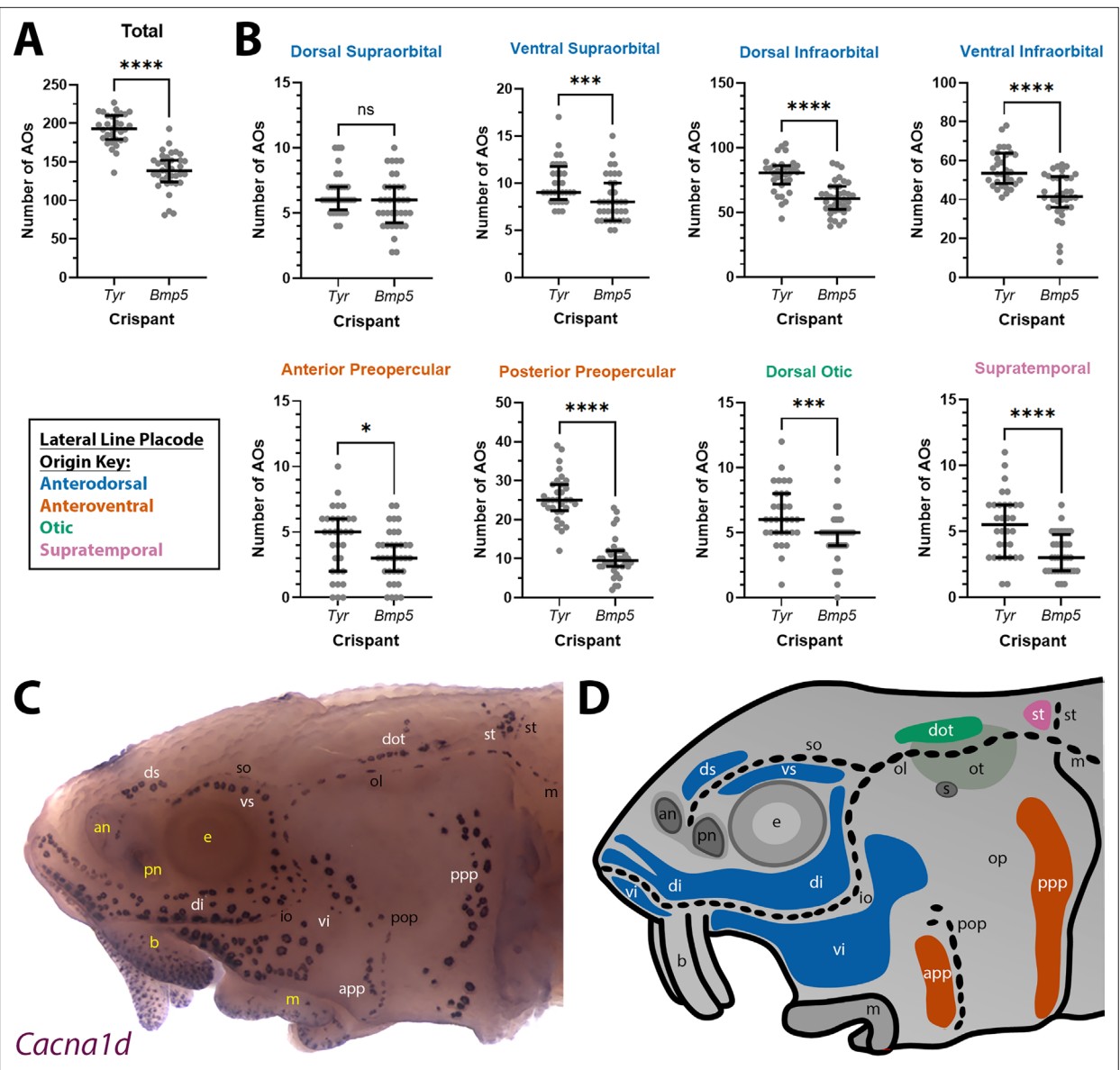

**Figure 5.** *Bmp5* crispants have significantly fewer ampullary organs than control *Tyr* crispants. (**A**) Scatter plot showing median and interquartile range for the total number of ampullary organs on one side of the head at stage 45 in *Bmp5* sterlet crispants (counted after in situ hybridisation [ISH] for *Cacna1d* or *Kcnab3*; n=36) versus control *Tyr* crispants (counted after ISH for *Cacna1d* or *Kcnab3*; n=32). *Bmp5* crispants have significantly fewer ampullary organs overall than control *Tyr* crispants (p<0.0001; two-tailed Mann-Whitney test). ***Supplementary file 2*** provides the sgRNA combination, injection batch and raw counts for each crispant. All the *Bmp5* crispants and 20 of the *Tyr* crispants used for statistical analysis were from the same batch. (**B**) Scatter plots showing median and interquartile range for the number of ampullary organs in each individual ampullary organ field on one side of the head at stage 45 in *Bmp5* crispants (n=36) versus control *Tyr* crispants (n=32). The raw counts are provided in ***Supplementary file 2***. For the location of each field, see panel **C** (*Cacna1d* expression) and panel **D** (schematic). Scatter plots are grouped with differently coloured titles according to lateral line placode (LLp) origin, following ***Gibbs and Northcutt, 2004***: blue, anterodorsal LLp (supraorbital and infraorbital fields); orange, anteroventral LLp (preopercular fields); green, otic LLp (dorsal otic field); pink, supratemporal LLp (supratemporal field). All fields except the dorsal supraorbital field have significantly fewer ampullary organs in *Bmp5* crispants versus control *Tyr* crispants (two-tailed Mann-Whitney tests). Symbols on plots represent p values: ns, not significant, p>0.05; *, p≤0.05; ***, p≤0.001; ****, p≤0.0001. Dorsal supraorbital: not significant, p=0.1207. Ventral supraorbital: p=0.0008. Dorsal infraorbital: p<0.0001. Ventral infraorbital: p<0.0001. Anterior preopercular: p=0.0466. Posterior preopercular: p<0.0001. Dorsal otic: p=0.0008. Supratemporal: p<0.0001. (**C**) Stage 45 sterlet head after ISH for the hair cell and electroreceptor marker *Cacna1d* (also expressed in taste buds on the barbels). Labels are white for ampullary organ fields; black for neuromast lines; yellow for anatomical landmarks. (**D**) Schematic of a stage 45 sterlet larval head. Ampullary organ fields are represented by coloured patches flanking the neuromast lines, which are represented as dotted lines. The different field colours indicate their lateral line placode origin (consistent with scatter plot titles in B). Abbreviations for ampullary organ fields: app, anterior preopercular; di, dorsal infraorbital; dot, dorsal otic; ds, dorsal supraorbital; ppp, posterior preopercular; st, supratemporal; vi, ventral infraorbital;

*Figure 5 continued on next page*

*Figure 5 continued*

vs, ventral supraorbital. Abbreviations for neuromast lines: io, infraorbital; m, middle; ol, otic; pop, preopercular; so, supraorbital; st, supratemporal. Abbreviations for anatomical landmarks: an, anterior naris; b, barbel; e, eye; m, mouth; op, operculum; ot, otic vesicle; pn, posterior naris; s, spiracle (first gill cleft).

sgRNAs contained either sgRNA 2 or sgRNA 4 (*Table 1*; *Supplementary file 1*) we expect all mixtures to have targeted both *Bmp5* ohnologs.

We targeted *Bmp5* using four different sgRNAs targeting exon 1 (*Table 1*; *Figure 4—figure supplement 1A*), injected in three different combinations of two to three different sgRNAs across two independent batches of one- to two-cell-stage embryos (*Supplementary file 1*). The *Bmp5*-targeted embryos (hereafter 'crispants') were raised to stage 45 (the onset of independent feeding, approximately 14 days post-fertilisation, dpf). ISH for the hair cell/electroreceptor marker *Cacna1d* (*Modrell et al., 2017a*; *Minařík et al., 2024a*) was used to visualise mature neuromasts and ampullary organs (i.e., differentiated hair cells and electroreceptors). Ampullary organ numbers in different fields vary considerably across individual larvae even in wild-types, but relative to *Tyr* crispants (n=0/24; *Figure 4A and B*; *Supplementary file 1*), we observed a mosaic reduction in *Cacna1d* expression in ampullary organ fields in 46% of *Bmp5* crispants (n=53/116; *Figure 4C–F*; *Supplementary file 1*). The efficacy of different sgRNA combinations varied significantly: injecting sgRNAs 2,3 led to fewer ampullary organs in 78% of cases (n=28/36; *Supplementary file 1*) versus 39% for sgRNAs 1,2,3 (n=16/41) and 23% for sgRNAs 1,4 (n=9/39; *Supplementary file 1*).

To confirm that our sgRNAs targeted the *Bmp5* locus, we genotyped 43 of the phenotypic *Bmp5* crispants by amplifying the sgRNA-targeted region from trunk/tail genomic DNA by PCR for direct Sanger sequencing. The nature and frequency of edits were analysed by subjecting the Sanger sequence data to in silico analysis using Synthego's online 'Inference of CRISPR Edits' (ICE) tool (*Conant et al., 2022*; also see, for example, *Uribe-Salazar et al., 2022*). *Figure 4—figure supplement 1B* shows a control *Tyr* crispant after ISH for *Cacna1d*, for comparison with two of the genotyped *Bmp5* crispants (*Figure 4—figure supplement 1C and D*). *Figure 4—figure supplement 1E–I* show examples of ICE output data revealing successful disruption of *Bmp5*; *Supplementary file 2* shows the ICE scores for each crispant analysed. Of the 43 genotyped crispants, 33 had a positive 'knock-out' score, confirming successful disruption of the targeted gene (*Supplementary file 2*). Our genotyping primers were designed before chromosome-level sterlet genomes were available; comparison with the reference genome (NCBI RefSeq assembly GCF_902713425.1) showed that only the chromosome 6 ohnolog can be amplified, owing to mismatches with the chromosome 5 ohnolog (primarily because the reverse primer targeted an intron). This, combined with crispant mosaicism, may explain why the ICE knock-out score was zero for ten *Bmp5* crispants that nevertheless displayed the phenotype of reduced number of ampullary organs.

To examine the disruption in ampullary organ formation further, 15 control *Tyr* crispants and 13 *Bmp5* crispants (six injected with sgRNAs 2,3 and seven injected with sgRNAs 1,4; *Supplementary file 1*) were subjected to ISH for the electroreceptor-specific marker *Kcnab3* (*Modrell et al., 2017a*; *Minařík et al., 2024a*). Relative to control *Tyr* crispants (*Figure 4G–I*), this confirmed the reduction in ampullary organ number when there was no possibility of confusing the two sensory organ types (*Figure 4J–L*). The same *Bmp5* crispants were then immunostained post-ISH for the supporting cell marker Sox2 (*Hernández et al., 2007*; *Modrell et al., 2017a*), which labels neuromasts more strongly than ampullary organs (*Modrell et al., 2017a*; *Minařík et al., 2024a*) and revealed no obvious phenotype in the number and morphology of neuromasts (*Figure 4J[1], K[1] and L[1]*; compare with *Figure 4J, K and L*). Furthermore, very few 'additional' ampullary organs appeared after Sox2 immunostaining (*Figure 4J[1], K[1] and L[1]*; compare with *Figure 4J, K and L*), suggesting that disrupting the *Bmp5* gene prevented ampullary organ formation, rather than blocking the later differentiation of *Kcnab3*-positive electroreceptors within ampullary organs.

Given the normal variation seen in ampullary organ number in different fields across individual larvae, we wished to test whether the qualitative phenotype of reduced ampullary organ number was statistically significant. We counted all the ampullary organs in each of the eight different fields on one side of the head of 36 phenotypic *Bmp5* crispants and 32 control *Tyr* crispants after ISH for *Cacna1d* or *Kcnab3*. The raw counting data are provided in *Supplementary file 2*. Statistical analysis using a two-tailed Mann-Whitney (Wilcoxon rank sum) test revealed that *Bmp5* crispants had significantly

fewer ampullary organs overall than *Tyr* control crispants (p<0.0001; *Figure 5A*). Indeed, all ampullary organ fields except for the dorsal supraorbital field (one of the smaller fields) had significantly fewer ampullary organs in *Bmp5* crispants versus control *Tyr* crispants (*Figure 5B*). *Figure 5C and D* show the location of each of the ampullary organ fields; the colour-coded schematic in *Figure 5D* also identifies their different lateral line placode origins (based on *Gibbs and Northcutt, 2004*). The dorsal supraorbital field originates from the anterodorsal lateral line placode, which also gives rise to the ventral supraorbital and the dorsal and ventral infraorbital fields, all of which had significantly fewer ampullary organs in *Bmp5* crispants versus control *Tyr* crispants (p=0.0008, p<0.0001 and p<0.0001, respectively; two-tailed Mann-Whitney test; *Figure 5B*). Thus, the lack of effect in the dorsal supraorbital field may simply reflect the relatively small number of ampullary organs (although this is not the smallest field).

Overall, these data show that CRISPR/Cas9-mediated targeting of *Bmp5* in G0-injected embryos led to significantly fewer ampullary organs developing in almost all fields. This suggests that *Bmp5*, which is expressed in ampullary organ primordia as well as in mature ampullary organs (*Figure 1*), normally acts to promote ampullary organ formation.

## Blocking Bmp signalling prior to ampullary organ formation results in supernumerary and ectopic ampullary organs

To explore the effect on ampullary organ development of blocking the Bmp pathway more generally than disrupting a specific ligand gene, we used a highly selective small-molecule Bmp inhibitor, DMH1 (dorsomorphin homolog 1; *Hao et al., 2010*; *Cross et al., 2011*). We treated stage 36 (newly hatched) sterlet yolk-sac larvae with DMH1 for 20 hours, by which point (at 16 °C) they will have reached approximately stage 38: that is, just prior to the onset of ampullary organ development (ISH for *Eya4* showed that ampullary organ primordia are present in all the main fields by stage 39; *Minařík et al., 2024a*). In comparison to DMSO controls (n=12), more ampullary organs had formed by stage 45 in all DMH1-treated larvae (n=17/17), as visualised by ISH for the hair cell/electroreceptor marker *Cacna1d* (*Figure 6A–D*; n=8) or for electroreceptor-specific *Kcnab3* (*Figure 6E–H*; n=9).

The increase in ampullary organ number seemed to be general but was most striking in the dorsalmost ampullary organ fields, i.e., the dorsal supraorbital, dorsal otic and supratemporal fields. These fields were clearly separate in the DMSO control larvae (*Figure 6A, B, E and F*). However, in the DMH1-treated larvae, there were so many ampullary organs that the fields appeared to fuse together in a line (*Figure 6C, D, G and H*). ISH for *Cacna1d* showed that the three dorsal ampullary organ fields were still clearly separate even in much older larvae (*Figure 6I and J*), suggesting that the supernumerary ampullary organs in this region of DMH1-treated larvae at stage 45 were ectopic, rather than precocious.

Although the increased number of ampullary organs in the dorsal fields was the most obvious and consistent phenotype, the *Cacna1d* expression pattern in several larvae suggested the presence of ectopic offshoots of the supraorbital neuromast line (*Figure 6K and L*; n=5/8). Initially, we could not determine from *Cacna1d* expression alone whether the ectopic organs were neuromasts or small ampullary organs, as this gene is expressed by both hair cells and electroreceptors. We therefore took six of the nine larvae that had been subjected to ISH for electroreceptor-specific *Kcnab3* and immunostained them for the supporting cell marker Sox2, which labels neuromasts more strongly than ampullary organs (also see *Minařík et al., 2024a*). This enabled direct comparison of the same larvae with and without visible neuromasts and showed that the ectopic organs were indeed neuromasts (*Figure 6M-N[1]*; n=5/6 as in one larva it was not clear whether an ectopic offshoot was indeed present).

Overall, therefore, ectopic offshoots of the supraorbital neuromast line (compare *Figure 6M[1],N[1]* with wildtype Sox2 expression at stage 45 in *Figure 2—figure supplement 1K and L*) were seen in a majority of larvae (n=10/14; 71%) in which Bmp signalling had been blocked for 20 hr from stage 36, where this could be determined (n=5/8 after ISH for *Cacna1d*; n=5/6 after ISH for *Kcnab3* followed by immunostaining for Sox2). At stage 36, neuromast primordia are already forming in the supraorbital primordium (as shown by Sox2 expression; *Figure 2—figure supplement 1E and F*). At stages 36–38, immunoreactivity for pSMAD1/5/9 suggests that Bmp signalling is most prominent in this region in lateral line nerves, rather than the supraorbital lateral line primordium (*Figure 2E–H*). This suggests the intriguing hypothesis that the Bmp signalling activity during stages 36–38 (the approximate period of DMH1 treatment) that is required to prevent ectopic secondary neuromast formation from the

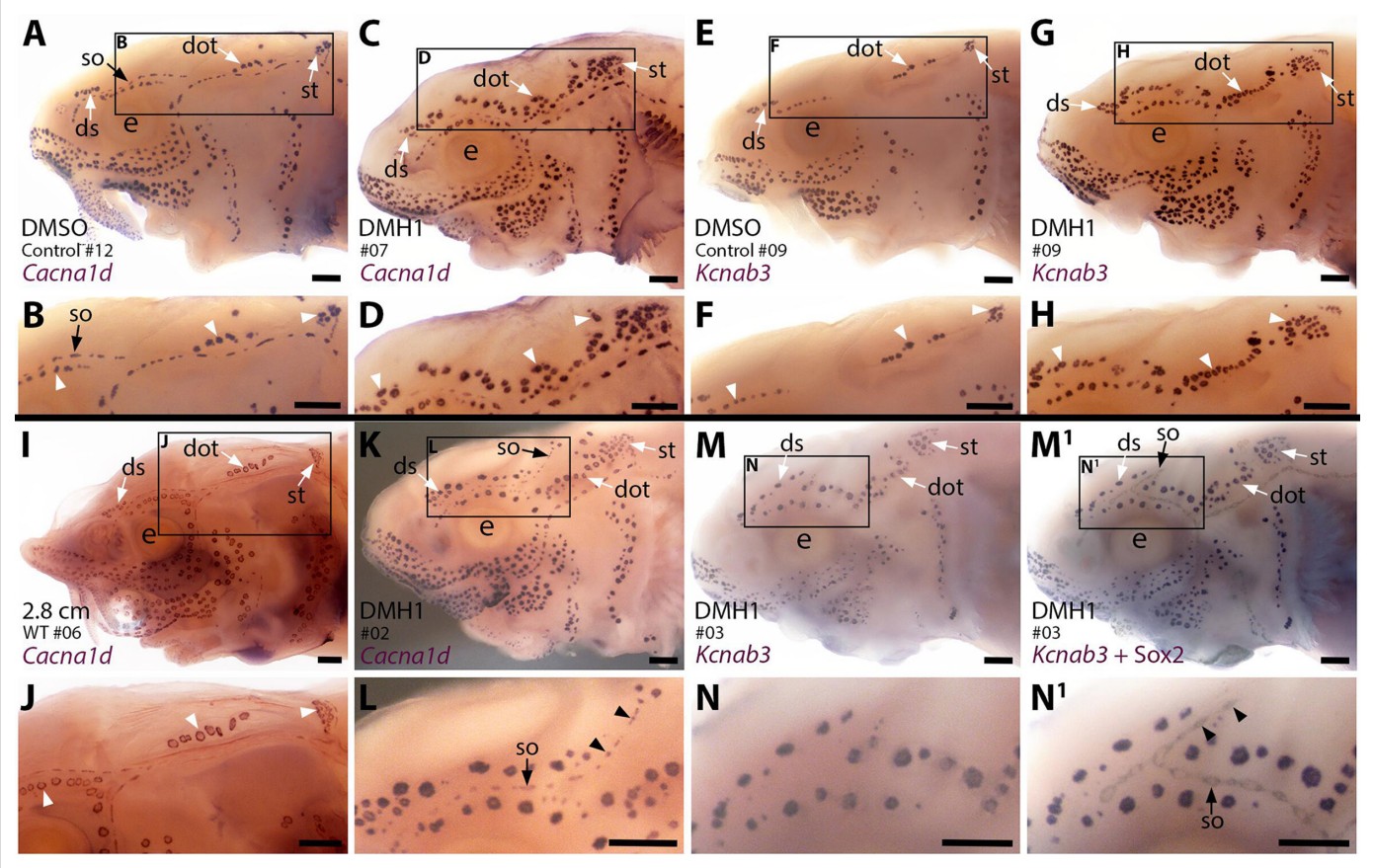

**Figure 6.** Sterlet larvae in which Bmp signalling was blocked prior to ampullary organ formation have supernumerary ampullary organs and ectopic supraorbital neuromasts. Sterlet larvae after in situ hybridisation (ISH) for the hair cell and electroreceptor marker *Cacna1d* (also expressed in taste buds on barbels) or the electroreceptor-specific marker *Kcnab3*. Black arrowheads indicate examples of neuromasts; white arrowheads indicate examples of ampullary organs. (**A–H**) Stage 45 larvae that had been treated for 20 hr from stage 36 (i.e., from hatching to approximately stage 38, just prior to the onset of ampullary organ development) with either DMH1 or DMSO as controls. Larvae are numbered for cross-referencing with ampullary organ counts in **Supplementary file 3**. ISH for *Cacna1d* (**A–D**) or *Kcnab3* (**E–H**) shows that, relative to DMSO-treated controls (**A,B,E,F**), DMH1-treated larvae have many more ampullary organs (**C,D,G,H**). This phenotype is particularly prominent in the three dorsal-most ampullary organ fields, where the dorsal supraorbital, dorsal otic and supratemporal fields - clearly separate in DMSO-treated larvae (**A,B,E,F**) - almost fuse together in DMH1-treated larvae (**C,D,G,H**). (**I,J**) A much older wild-type larva (2.8 cm in length, ~65 dpf) after ISH for *Cacna1d*. The dorsal supraorbital, dorsal otic and supratemporal ampullary organ fields are clearly separated, suggesting the supernumerary ampullary organs in this region in DMH1-treated larvae (**C,D,G,H**) are ectopic, not precocious. (**K–N¹**) Most DMH1-treated larvae also develop an ectopic offshoot from the supraorbital neuromast line. This is visible after ISH for *Cacna1d* (**K,L**; compare with DMSO control in **A,B**) and confirmed to represent neuromasts in DMH1-treated larvae via ISH for electroreceptor-specific *Kcnab3* (**M,N**) followed by immunostaining for the supporting cell marker Sox2 to reveal neuromasts (**M¹,N¹**). Abbreviations: dot, dorsal otic ampullary organ field; ds, dorsal supraorbital ampullary organ field; e, eye; S, stage; so, supraorbital neuromast line; st, supratemporal ampullary organ field; WT, wild type. Scale bar: 250 µm.

supraorbital neuromast line, might be active in lateral line nerves, rather than the lateral line primordium itself.

Finally, given the normal variation seen in ampullary organ number in different fields across individual larvae, we wished to test whether the qualitative phenotype of increased ampullary organ number at stage 45 in DMH1-treated versus DMSO control larvae was statistically significant. This included in other ampullary organ fields besides the dorsalmost fields where supernumerary, ectopic ampullary organs were obvious (**Figure 6A–K**). We therefore counted all the ampullary organs in each of the eight different fields on one side of the head of the stage 45 DMH1-treated larvae (n=17), stage 45 DMSO control larvae (n=12) and older wild-type larvae (either 2.0 cm or 2.8 cm in length, i.e., approximately 50 or 65 dpf; n=10). **Supplementary file 3** shows the raw counting data. Statistical analysis using a two-tailed Mann-Whitney (Wilcoxon rank sum) test confirmed that DMH1-treated larvae had significantly more ampullary organs overall than DMSO controls (p<0.0001; **Figure 7A**).

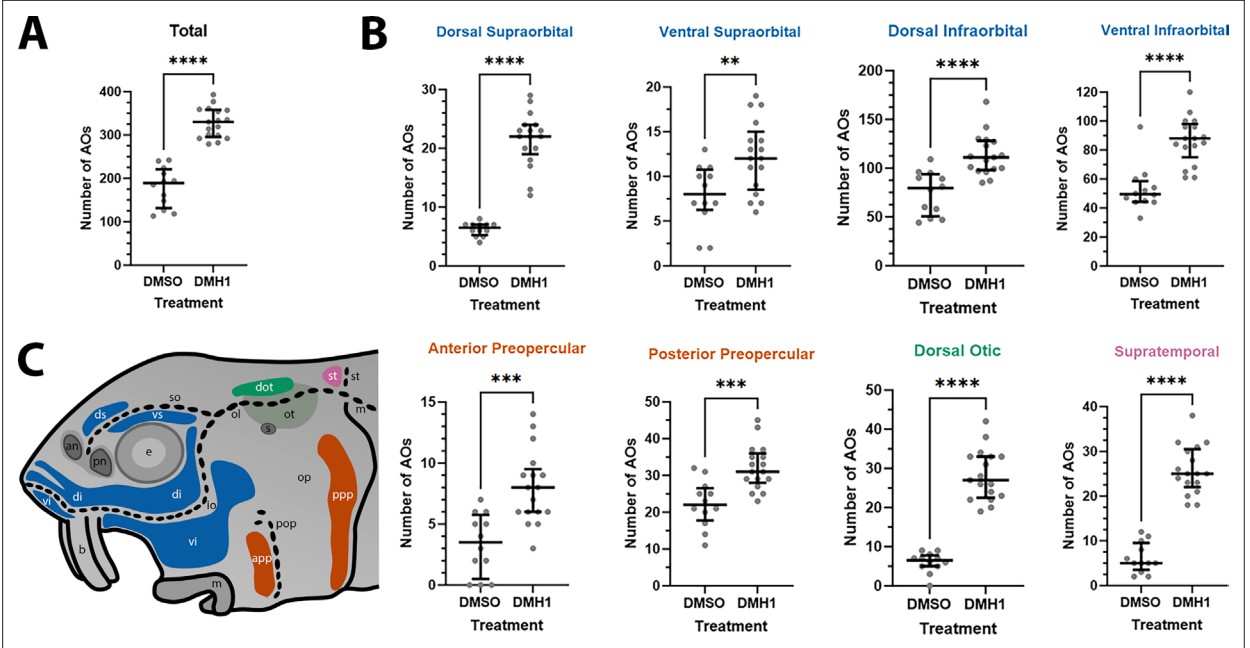

**Figure 7.** DMH1-treated larvae have significantly more ampullary organs than DMSO controls. (**A**) Scatter plot showing median and interquartile range for the total number of ampullary organs on one side of the head in stage 45 sterlet larvae that had been treated for 20 hours from stage 36 (i.e., from hatching to approximately stage 38, just prior to the onset of ampullary organ development) with DMH1 (n=17) or DMSO as controls (n=12). DMH1-treated larvae have significantly more ampullary organs (p<0.0001; two-tailed Mann-Whitney test). Ampullary organs were counted after in situ hybridisation [ISH] for *Cacna1d* or *Kcnab3*; raw counts are provided in **Supplementary file 3**. (**B**) Scatter plots showing median and interquartile range for the number of ampullary organs in each individual ampullary organ field on one side of the head in stage 45 sterlet larvae that had been treated for 20 hr from stage 36 with DMH1 (n=17), versus with DMSO as controls (n=12). Raw counts are provided in **Supplementary file 3**. For the location of each field, see schematic in panel **C** (reproduced from **Figure 5D**). Scatter plots are grouped with differently coloured titles according to lateral line placode (LLp) origin, following **Gibbs and Northcutt, 2004**: blue, anterodorsal LLp origin (supraorbital and infraorbital fields); orange, anteroventral LLp origin (preopercular fields); green, otic LLp origin (dorsal otic field); pink, supratemporal LLp origin (supratemporal field). All fields have significantly more ampullary organs in DMH1-treated larvae (n=17) than in DMSO controls (n=12; two-tailed Mann-Whitney tests). Asterisks on plots represent p values: **, p≤0.01; ***, p≤0.001; ****, p≤0.0001. p values for all fields are <0.0001 except for the ventral supraorbital field (p=0.0074), anterior preopercular field (p=0.0002) and posterior preopercular field (p=0.0003). (**C**) Schematic of a stage 45 sterlet larval head. Ampullary organ fields are represented by coloured patches flanking the neuromast lines, which are represented as dotted lines. The different field colours indicate their lateral line placode origin (consistent with scatter plot titles in **B**). Abbreviations for ampullary organ fields: app, anterior preopercular; di, dorsal infraorbital; dot, dorsal otic; ds, dorsal supraorbital; ppp, posterior preopercular; st, supratemporal; vi, ventral infraorbital; vs, ventral supraorbital. Abbreviations for neuromast lines: io, infraorbital; m, middle; ol, otic; pop, preopercular; so, supraorbital; st, supratemporal. Abbreviations for anatomical landmarks: an, anterior naris; b, barbel; e, eye; m, mouth; op, operculum; ot, otic vesicle; pn, posterior naris; s, spiracle (first gill cleft).

This was also the case for each individual ampullary organ field (**Figure 7B**; **Supplementary file 3**). The colour-coded schematic in **Figure 7C** shows the location of each field and their different lateral line placode origins (based on **Gibbs and Northcutt, 2004**).

Furthermore, the DMH1-treated larvae (n=17) had significantly more ampullary organs than older (2.0/2.8 cm) wild-type larvae (n=10), both overall (p<0.0001; **Figure 8A**) and in all fields except the ventral supraorbital and posterior preopercular fields (**Figure 8B**; **Supplementary file 3**; the same colour-coded schematic is shown in **Figure 8C**). (Note: the five 2.0 cm and five 2.8 cm wild-type larvae were grouped together for statistical comparison with DMH1-treated larvae because using a two-tailed Mann-Whitney test showed that there was no significant difference between ampullary organ numbers in 2.0 cm versus 2.8 cm larvae, either overall [p=0.4206] or in any individual field [p>0.05 for each field].)

Overall, these results show that blocking Bmp signalling for 20 hr from stage 36, before the first ampullary organ primordia form, results in supernumerary ampullary organs in all fields. Furthermore, ectopic ampullary organs form in the dorsalmost fields, and an ectopic offshoot of the supraorbital neuromast line develops in a majority of larvae. This suggests that during normal development, Bmp

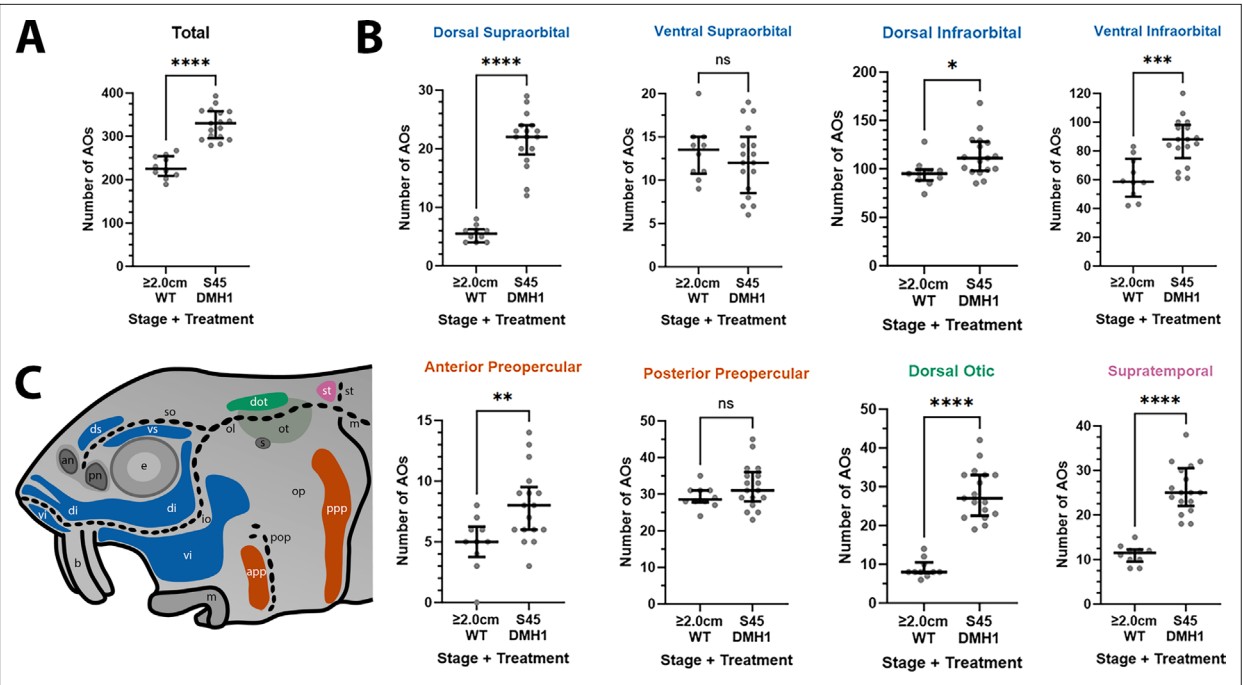

**Figure 8.** DMH1-treated larvae have significantly more ampullary organs at stage 45 than older wild-type larvae. (**A,B**) Scatter plots showing median and interquartile range for the number of ampullary organs on one side of the head in stage 45 sterlet larvae that had been treated for 20 hr from stage 36 with DMH1 (n=17) vs 2.0/2.8 cm wild-type larvae (~50/65 dpf; n=10). Raw counts are provided in **Supplementary file 3**. Two-tailed Mann-Whitney tests were used for statistical analysis. DMH1-treated larvae have significantly more ampullary organs overall at stage 45 than wild-type older larvae (p<0.0001; panel **A**). In panel **B**, scatter plots are grouped with differently coloured titles according to lateral line placode (LLp) origin, following **Gibbs and Northcutt, 2004**: blue, anterodorsal LLp (supraorbital and infraorbital fields); orange, anteroventral LLp (preopercular fields); green, otic LLp (dorsal otic field); pink, supratemporal LLp (supratemporal field). DMH1-treated larvae have significantly more ampullary organs at stage 45 than older wild-type larvae in all fields except the ventral supraorbital and posterior preopercular fields. Symbols on plots represent p values: ns, not significant, (p>0.05; *, p≤0.05; **, p≤0.01; ***, p≤0.001; ****, p≤0.0001). Dorsal supraorbital: p<0.0001. Ventral supraorbital: not significant (p=0.5109). Dorsal infraorbital: p=0.0123. Ventral infraorbital: p=0.0002. Anterior preopercular: p=0083. Posterior preopercular: not significant (p=0.1789). Dorsal otic: p<0.0001. Supratemporal: p<0.0001. (**C**) Schematic of a stage 45 sterlet larval head. Ampullary organ fields are represented by coloured patches flanking the neuromast lines, which are represented as dotted lines. The different field colours indicate their lateral line placode origin (consistent with scatter plot titles in **B**). Abbreviations for ampullary organ fields: app, anterior preopercular; di, dorsal infraorbital; dot, dorsal otic; ds, dorsal supraorbital; ppp, posterior preopercular; st, supratemporal; vi, ventral infraorbital; vs, ventral supraorbital. Abbreviations for neuromast lines: io, infraorbital; m, middle; ol, otic; pop, preopercular; so, supraorbital; st, supratemporal. Other abbreviations: an, anterior naris; b, barbel; e, eye; m, mouth; op, operculum; ot, otic vesicle; pn, posterior naris; s, spiracle (first gill cleft); WT, wild type.

signalling dampens ampullary organ formation, preventing the over-production of ampullary organs in each individual field and the formation of ectopic ampullary organs in the most dorsal fields.

## Discussion

In this study, we identified opposing roles for Bmp signalling in ampullary organ development in sterlet. We began by investigating *Bmp5*, the only Bmp ligand gene in our late-larval paddlefish lateral line organ-enriched gene-set (**Modrell et al., 2017a**). In sterlet, *Bmp5* proved to be expressed in ampullary organ primordia (though not neuromast primordia), as well as in mature ampullary organs and neuromasts. Significantly fewer ampullary organs formed when *Bmp5* was targeted for CRISPR/Cas9-mediated mutagenesis in G0-injected sterlet embryos, suggesting that during normal development, Bmp5 promotes ampullary organ formation. In contrast, blocking Bmp signalling globally at stages just prior to the onset of ampullary organ development led to significantly more ampullary organs forming in all fields. Hence, Bmp signalling activity is required to prevent too many ampullary organs from developing. Taken together, therefore, our study has uncovered opposing roles for Bmp signalling during ampullary organ formation.

## Bmp5 promotes ampullary organ formation in sterlet

The early expression of *Bmp5* in ampullary organ primordia, but not neuromast primordia, suggested a role specifically in ampullary organ development. Indeed, targeting *Bmp5* for CRISPR/Cas9-mediated mutagenesis in G0-injected sterlet embryos led to significantly fewer ampullary organs forming, with no effect on neuromast formation. Its precise function and timing of action in promoting ampullary organ formation remain to be determined. Although beyond the scope of this study, it could be informative to test whether inhibiting Bmp signalling using DMH1 for discrete periods at successively later time-points recapitulates the *Bmp5* crispant phenotype. Similarly, it would be interesting to see whether developing ampullary organs in *Bmp5* crispants show any changes in the expression of the Bmp/Wnt inhibitor genes *Sostdc1* and/or *Apcdd1*.

In zebrafish, *Bmp5* expression has been reported in the migrating posterior lateral line primordium (*Thisse and Thisse, 2004*), which also expresses *Bmp4b* and *Bmp2a* (*Mowbray et al., 2001*). However, a role for Bmp signalling in neuromast development has not been identified (see, for example, *Piotrowski and Baker, 2014*; *Chitnis, 2021*). Small-molecule inhibition of Bmp signalling from late epiboly or neural plate stages led to expansion of the posterior (but not pre-otic) lateral line placode, suggesting that a much earlier phase of Bmp signalling restricts the posterior lateral line placode from expanding both posteriorly and laterally (*Nikaido et al., 2017*).

*Bmp5* was also expressed at later stages in ampullary organs and transiently in neuromasts, after electroreceptors/hair cells have differentiated. In mature neuromasts in zebrafish (at 5 dpf), scRNA-seq data show that *Bmp5* is expressed in neuromast hair cell progenitor populations and downregulated as hair cells differentiate (*Lush et al., 2019*). Furthermore, *Bmp5* is among the genes upregulated in 5-dpf zebrafish neuromasts within one hour after neomycin-induced hair cell death (*Jiang et al., 2014*; *Heller et al., 2022*), and in the postnatal mouse cochlea after gentamycin-induced hair cell death (*Bai et al., 2019*). Hence, Bmp5 may be important for hair cell regeneration. Neomycin treatment at late-larval stages (stages 44/45) in the Siberian sturgeon (*A. baerii*) kills electroreceptors, as well as hair cells, both of which subsequently regenerate (*Fan et al., 2016*; *Wang et al., 2020*). Given the expression of *Bmp5* in mature ampullary organs and neuromasts in sterlet, Bmp5 could play a role in the homeostasis (and regeneration after injury) of electroreceptors as well as hair cells.

## Bmp signalling prevents supernumerary and ectopic ampullary organs from forming

In addition to *Bmp5* expression in developing ampullary organ primordia (but not neuromast primordia) and mature ampullary organs and neuromasts, we also identified diffuse, more transient *Bmp4* expression between stages 40–42 within developing ampullary organ fields and neuromast regions. Persistent expression was also seen in the migrating lateral line primordia on the trunk, consistent with a report of *Bmp4b* (as well as *Bmp2a*) expression in the migrating posterior lateral line primordium (priml) in zebrafish (*Mowbray et al., 2001*). Additional unidentified Bmp ligand(s) are also likely to be expressed in sterlet, as pSMAD1/5/9 immunoreactivity (a proxy for Bmp signalling pathway activity; *Schmierer and Hill, 2007*) was seen throughout lateral line development, including within elongating lateral line primordia and afferent lateral line nerves (which extend together with all lateral line primordia as they elongate or migrate; *Winklbauer, 1989*; *Northcutt, 2005*; *Piotrowski and Baker, 2014*), as well as at the periphery of developing ampullary organs and neuromasts.

We identified a role for Bmp signalling in preventing too many ampullary organs from forming, using the selective Bmp pathway inhibitor DMH1 (*Hao et al., 2010*). DMH1 blocks signalling through the type I receptors Acvr1 (Alk2), Acvrl1 (Alk1), and Bmpr1a (Alk3) (*Hao et al., 2010*; *Cross et al., 2011*), all of which signal via Smad1/5/9 (*Yadin et al., 2016*). Acvr1 (Alk2) binds Bmp5/6/7/8; Acvrl1 (Alk1) binds Bmp9/10, and Bmpr1a (Alk3) binds Bmp2/4/5/6/7/8 and Gdf5/6/7 (also known as Bmp14/13/12) (*Yadin et al., 2016*). We blocked Bmp signalling globally in sterlet yolk-sac larvae just before the onset of ampullary organ development, by treating them with DMH1 for 20 hr from stage 36 (hatching) to approximately stage 38. By the onset of independent feeding at stage 45, significantly more ampullary organs had formed in all fields relative to DMSO controls, and ectopic ampullary organs had formed in the three dorsalmost fields (the dorsal supraorbital, dorsal otic and supratemporal fields), in regions where ampullary organs are not seen even in much older post-feeding larvae. This suggests Bmp signalling normally prevents supernumerary and ectopic ampullary organs from forming.

Although a role for Bmp signalling has not been identified in neuromast formation (see, for example, *Piotrowski and Baker, 2014*; *Chitnis, 2021*), this pathway is important for the formation of inner-ear sensory patches, within which hair cells also differentiate. *Bmp4* is an early marker for all sensory patches in the chicken inner ear, and for the cristae (vestibular sensory patches of the semicircular canals) in mouse (*Wu and Oh, 1996*; *Morsli et al., 1998*). Conditional knockout experiments showed that *Bmp4* is required for the formation of the cristae (*Chang et al., 2008*). *Bmp4* is also expressed in the developing cochlea, and conditional knockout of the type I receptor genes *Bmpr1a* (*Alk3*) and *Bmpr1b* (*Alk6*) showed that Bmp signalling is also required for the induction of the cochlear-duct prosensory domain that forms the organ of Corti (*Ohyama et al., 2010*).

Treatment of cultured mouse otocysts with different concentrations of Bmp4 revealed that intermediate levels of Bmp4 promote hair cell formation (*Ohyama et al., 2010*). Conflicting results were reported from Bmp4 treatment of chicken otocysts explanted at embryonic days 3–4: this either increased hair cell number (*Li et al., 2005*) or reduced the size of *Atoh1*-positive sensory patches and increased cell death (*Pujades et al., 2006*). *Ohyama et al., 2010* suggested that the differences seen could reflect the concentrations of Bmp4 used being lower (hair cell-promoting; *Li et al., 2005*) versus higher (hair cell-inhibiting; *Pujades et al., 2006*). A subsequent study of developing chicken cristae found that both *Bmp4* expression and pSmad1/5/9 immunoreactivity (a proxy for Bmp signalling) were high in most cells of the cristae except in differentiating hair cells, where both were downregulated (*Kamaid et al., 2010*). In contrast, in the mature (post-hatching) chicken auditory epithelium (basilar papilla), *Bmp4* was highly expressed in hair cells but not supporting cells, and type I receptor genes (*Bmpr1a*, *Bmpr1b*) and a type II receptor gene (*Bmpr2*) were expressed in both hair cells and supporting cells (*Lewis et al., 2018*). After killing hair cells by treating explanted basilar papilla with aminoglycoside antibiotics, supporting cells differentiated into hair cells (either after proliferating or directly via transdifferentiation), and *Bmp4* was also expressed in such regenerated hair cells (*Lewis et al., 2018*). Application of Bmp4 with the ototoxic antibiotic blocked hair cell regeneration by preventing supporting cells from proliferating and upregulating *Atoh1* (*Lewis et al., 2018*). Conversely, application of the extracellular Bmp4/2/7 antagonist Noggin (*Zimmerman et al., 1996*) together with the ototoxic antibiotic led to the formation of significantly more hair cells per unit area than in control cultures (*Lewis et al., 2018*). Taken together, these results suggest that in the mature auditory epithelium, Bmp4 secreted from existing hair cells prevents supporting cells from forming supernumerary hair cells; after hair-cell death, Bmp4 is lost and this inhibition is relieved, allowing hair cell regeneration (*Lewis et al., 2018*).

The regeneration of supernumerary hair cells in the mature chicken auditory epithelium after inhibiting Bmp signalling with Noggin (*Lewis et al., 2018*) was reminiscent of the formation of supernumerary ampullary organs after inhibiting Bmp signalling with DMH1, prior to the onset of ampullary organ development. Indeed, we also note the action of Bmps as inhibitors in reaction-diffusion (Turing) systems (see *Green and Sharpe, 2015*) that result in the periodic spacing of hair follicles (*Mou et al., 2006*), feather primordia (*Jung et al., 1998*; *Noramly and Morgan, 1998*; *Jiang et al., 1999*; *Michon et al., 2008*) and potentially also denticles in shark skin (*Cooper et al., 2018*). Sterlet *Bmp4* was expressed in the regions where ampullary organs and neuromasts are forming on the head (and more strongly in the migrating lateral line primordia on the trunk), but only weakly and transiently in developing ampullary organs and neuromasts themselves. This could be consistent with a role for Bmp4 in promoting formation of the prosensory domain within which the sensory organs develop, as seen for inner ear sensory patches (*Chang et al., 2008*; *Ohyama et al., 2010*). Its subsequent downregulation in developing lateral line organs in sterlet differs from the expression of chicken *Bmp4* in the vestibular cristae (*Bmp4*-positive supporting cells; *Kamaid et al., 2010*) and auditory basilar papilla (*Bmp4*-positive hair cells; *Lewis et al., 2018*). However, *Bmp5* is expressed in mature ampullary organs and neuromasts and additional as-yet unidentified Bmp ligand genes may also be expressed, given the more extensive pattern of pSMAD1/5/9 immunoreactivity. Furthermore, the dual Bmp/Wnt inhibitor genes *Sostdc1* and *Apcdd1* are both expressed during ampullary organ development: their roles are unknown, and the expression of one or both genes could be regulated by Bmp signalling. Overall, the precise mechanism by which Bmp signalling normally prevents supernumerary and ectopic ampullary organ formation remains to be established, but the data from the chicken auditory epithelium (*Lewis et al., 2018*) and reaction-diffusion systems patterning other skin structures (see *Green and Sharpe, 2015*) provide potential parallels for future investigation.

We recently reported that the transcription factor gene *Foxg1* is expressed in paddlefish and sterlet in the central region of sensory ridges where neuromasts form (*Minařík et al., 2024a*), and that targeting *Foxg1* for CRISPR/Cas9-mediated mutagenesis led to ampullary organs forming within neuromast lines (preprint, *Minařík et al., 2024b*). Here, we found that Bmp signalling is required to prevent supernumerary ampullary organ formation within ampullary organ fields, including ectopic ampullary organs within the small dorsalmost fields, although neuromast lines developed normally (apart from the ectopic offshoot of the supraorbital line; see next section). Although these phenotypes are distinct, a common theme emerges, namely the active repression of ampullary organ formation during normal development: within neuromast lines by Foxg1 (whether directly or indirectly), and within ampullary organ fields by Bmp signalling. Taken together, this suggests that lateral line primordia are 'poised' to form ampullary organs (indeed potentially that ampullary organs are the 'default' fate for lateral line primordia in electroreceptive species; preprint, *Minařík et al., 2024b*), and this must be controlled to ensure that ampullary organs develop in the 'correct' number and location.

## Bmp signalling activity prevents ectopic secondary neuromast formation in the supraorbital neuromast line

An ectopic offshoot of the supraorbital neuromast line also developed by stage 45 in a majority of larvae that had been treated with DMH1 to block Bmp signalling for 20 hr from hatching (stages 36–38). Intriguingly, pSMAD1/5/9 immunoreactivity (a proxy for Bmp signalling activity) was particularly prominent within lateral line nerves from stages 36–40, including the supraorbital nerve (nerve immunoreactivity had almost disappeared by stage 42), and was also prominent in the supraorbital region at later stages. Afferent innervation is not required for the formation of neuromasts deposited by lateral line primordia in zebrafish (*Andermann et al., 2002*; *Grant et al., 2005*; *López-Schier and Hudspeth, 2005*). However, the post-embryonic budding of neuromasts to form short rows ('stitches') of additional neuromasts depends on Wnt signalling from afferent axons: this promotes cell proliferation within the neuromast, which is required for the budding process (*Wada et al., 2013*; *Wada and Kawakami, 2015*). We speculate that Bmp signalling in the lateral line nerve may act to inhibit this process during embryogenesis, thus preventing precocious budding of primary neuromasts. This hypothesis remains to be tested.

## Conclusion

Overall, we have identified opposing roles for Bmp signalling during the development of electrosensory ampullary organs in the sterlet. CRISPR/Cas9-mediated mutagenesis in G0-injected embryos showed that *Bmp5*, which is expressed within ampullary organ primordia (and later in mature ampullary organs and neuromasts), is required for ampullary organ formation. Conversely, global inhibition of type I Bmp receptors via DMH1 treatment at stages just prior to the onset of ampullary organ development, revealed that Bmp signalling is required to prevent supernumerary and ectopic ampullary organs from forming. Future work will be required to understand the respective mechanisms involved.

# Materials and methods
## Collection, staging and fixation of sterlet embryos and larvae

Fertilised sterlet (*Acipenser ruthenus*) eggs were obtained during the annual spawning season at the Research Institute of Fish Culture and Hydrobiology (RIFCH), Faculty of Fisheries and Protection of Waters, University of South Bohemia in České Budějovice (Vodňany, Czech Republic). Comprehensive information about sterlet husbandry, in vitro fertilisation and the rearing of embryos and yolk-sac larvae is provided by *Stundl et al., 2022*. A mix of sperm from three different males was used for each fertilisation, so each batch comprised siblings and half-siblings. At desired stages (*Dettlaff et al., 1993*), embryos/larvae were euthanised by anaesthetic overdose using MS-222 (Sigma-Aldrich) before fixation in modified Carnoy's fixative (6 volumes 100% ethanol: 3 volumes 37% formaldehyde: 1 volume glacial acetic acid) for 3 hr at room temperature and graded into ethanol for storage at –20 °C.

All experimental procedures were approved by the Animal Research Committee of the Faculty of Fisheries and Protection of Waters in Vodňany, University of South Bohemia in České Budějovice, Czech Republic, and by the Ministry of Agriculture of the Czech Republic (reference number: MSMT-12550/2016–3). Experimental fish were maintained according to the principles of the European Union (EU) Harmonized Animal Welfare Act of the Czech Republic, and Principles of Laboratory Animal Care and National Laws 246/1992 'Animal Welfare' on the protection of animals.

## Gene cloning, in situ hybridisation and immunohistochemistry

Total RNA was extracted from the heads of stage 45 sterlet larvae using Trizol (Invitrogen, Thermo Fisher Scientific), treated with DNAse using the Ambion Turbo DNA-free kit (Invitrogen, Thermo Fisher Scientific) and cDNA synthesised using the High-Capacity cDNA Reverse Transcription Kit (Applied Biosystems), following the manufacturers' instructions. Genes were selected from the late-larval paddlefish (*Polyodon spathula*) lateral line organ-enriched gene-set (National Center for Biotechnology Information [NCBI] Gene Expression Omnibus accession code GSE92470; *Modrell et al., 2017a*) or via a candidate approach. The relevant paddlefish transcriptome sequence was used in a command-line search of a Basic Local Alignment Search Tool (BLAST) database generated from our sterlet transcriptome assemblies (from pooled late-larval sterlet heads at stages 40–45; *Minařík et al., 2024a*), which are available at DDBJ/EMBL/GenBank under the accessions GKLU00000000 and GKEF01000000. Sterlet sequence identity was confirmed using NCBI BLAST (https://blast.ncbi.nlm.nih.gov/Blast.cgi; *McGinnis and Madden, 2004*). PCR primers (*Supplementary file 4*) were designed using Primer3Plus (*Untergasser et al., 2012*), which is also integrated into Benchling's Editor program (https://benchling.com), and used under standard PCR conditions to amplify cDNA fragments from sterlet cDNA. These were cloned into QIAGEN's pDrive cloning vector using the QIAGEN PCR Cloning Kit (QIAGEN) and clones verified by sequencing (Department of Biochemistry Sequencing Facility, University of Cambridge). Sequence identity was confirmed using NCBI BLAST. Alternatively, sterlet transcriptome data were used to design synthetic gene fragments with added M13 forward and reverse primer adaptors, which were purchased from Twist Bioscience.

Chromosome-level genome assemblies for sterlet (*Du et al., 2020* and the 2022 reference genome, NCBI Refseq assembly GCF_902713425.1/) had not been published when these sterlet riboprobe template sequences were designed. Both ohnologs (gene paralogs resulting from whole-genome duplication) for all genes described here have been retained from the independent whole-genome duplication in the sterlet lineage (*Du et al., 2020*). *Supplementary file 4* includes each riboprobe's percentage match with each ohnolog, obtained by using NCBI BLAST to perform a nucleotide BLAST search against the respective genome assemblies. The percentage match with the 'targeted' ohnolog ranged from 99.2 to 100%. The percentage match with the second ohnolog ranged from 90.0 to 100%, suggesting that our riboprobes most likely also target transcripts from the second ohnolog. GenBank accession numbers for the top match for each riboprobe, and the nucleotide ranges targeted, are given in *Supplementary file 4*.

Digoxigenin-labelled riboprobes were synthesised as previously described (*Minařík et al., 2024a*). Wholemount in situ hybridisation (ISH) was performed as described in *Modrell et al., 2011a*. Wholemount immunostaining was performed as described in *Metscher and Müller, 2011*. Primary antibodies (anti-Sox2: Abcam ab92494, rabbit monoclonal, 1:200; anti-Phospho-SMAD1/5/9: Cell Signalling Technology D5B10, rabbit monoclonal, 1:100) were applied in blocking solution for 24 hr at 4 °C, as was the secondary antibody (horseradish peroxidase-conjugated goat anti-rabbit IgG: Jackson ImmunoResearch, 1:500). The metallographic peroxidase substrate EnzMet kit (Nanoprobes 6010) was used for the colour reaction, following the manufacturer's instructions. For both ISH and immunostaining, at least three embryos/larvae were used per stage.

## CRISPR guide RNA design and synthesis

Prior to the publication of chromosome-level sterlet genomes (*Du et al., 2020* and the 2022 NCBI RefSeq assembly GCF_902713425.1), *Bmp5* was identified using NCBI BLAST to search draft genomic sequence data (M.H., unpublished). Exons were confirmed by comparison with spotted gar (*Lepisosteus oculatus*) using Ensembl (*Cunningham et al., 2022*). NCBI BLASTX (https://blast.ncbi.nlm.nih.gov/Blast.cgi; *McGinnis and Madden, 2004*) was used to identify conserved domains. Four CRISPR single guide RNAs (sgRNAs) were designed using the CRISPR Guide RNA Design Tool from Benchling

([https://benchling.com](https://benchling.com)) to target a 450-base region within exon 1 that encodes part of the TGFβ propeptide domain (*Table 1*; *Figure 4—figure supplement 1A*). The previously published guides against *tyrosinase* were designed as described in *Minařík et al., 2024b*, preprint.

Plasmid pX335-U6-Chimeric_BB-CBh-hSpCas9n(D10A) (Addgene, plasmid #42335; *Cong et al., 2013*) was used to synthesize DNA templates containing the single guide (sg)RNA scaffold, which was amplified using the same reverse primer for all reactions (AAAAAAGCACCGACTCGGTGCC; personal communication, Dr Ahmed Elewa, Karolinska Institutet, Stockholm, Sweden) and a specific forward primer for each sgRNA. The forward primer had an overhang containing the T7 promoter and the 20-nucleotide sgRNA target sequence: GATCAC<u>TAATACGACTCACTATA</u>(20N)GTTTTAGA GCTAGAAAT, where the T7 promoter is underlined and "(20N)" represents the target sequence specific to each sgRNA (*Table 1*). Where the first nucleotide of the target sequence was G, this completed the T7 promoter (and became the first base of the sgRNA). Where the target sequence did not start with G, an additional G was added before the target sequence to complete the T7 promoter and ensure efficient transcription. Q5 DNA polymerase (New England Biolabs, NEB) was used to amplify the DNA template, which was purified using the Monarch PCR & DNA Cleanup Kit (NEB). The HiScribe T7 High Yield RNA Synthesis Kit (NEB) was used to synthesise the sgRNAs, which were purified using the Monarch RNA Cleanup Kit (NEB) and stored at −80 °C. Alternatively, chemically modified synthetic gRNAs were purchased from Synthego (CRISPRevolution sgRNA EZ Kit).

## Embryo injections and genotyping

A detailed description of sterlet embryo injection is provided in *Minařík et al., 2024b*, preprint. Briefly, 2400 ng Cas9 protein with NLS (PNA Bio CP01) were combined with 1200 ng of sgRNA in 4.5 µl nuclease-free water on the day of injection and left at room temperature for 10 min to form ribonucleoprotein complexes, then kept on ice. For sgRNA multiplexing, different Cas9-sgRNA complexes were combined 1:1 and 0.5 µl 10% 10,000 MW rhodamine dextran (Invitrogen) added to a final volume of 5 µl. One- or two-cell-stage embryos were injected with approximately 20 nl of the injection mixture (manually or using an Eppendorf FemtoJet 4 x microinjector) and maintained at 20 °C until the 64-cell stage, then transferred to 16 °C. Upon reaching stage 45, they were euthanised by MS-222 overdose, fixed with modified Carnoy's fixative and dehydrated into ethanol as described above. Prior to ISH, fixed crispants were cut in half and the tails set aside for genotyping. DNA was extracted from crispant tails using the PCRBIO Rapid Extract PCR Kit (PCR Biosystems) and the target region amplified using HS Taq Mix Red (PCR Biosystems) following the manufacturer's instructions. Genotyping primers (*Supplementary file 1*) were designed using Benchling's Editor program ([https://benchling.com](https://benchling.com)) to flank the sgRNA target region with a buffer of at least 150 bp. PCR products were subjected to agarose gel electrophoresis, extracted using the MinElute Gel Extraction Kit (QIAGEN) according to the manufacturer's protocol and sequenced by Genewiz (Azenta Life Sciences). The resulting Sanger trace files were uploaded for analysis by Synthego's Inference of CRISPR Edits (ICE) tool (*Conant et al., 2022*).

## Small-molecule inhibition of Bmp signalling

Stage 36 (post-hatching) yolk-sac larvae were incubated for 20 hr in 50 µM DMH1 (Cayman Chemical) in 1% dimethyl sulfoxide (DMSO) or in 1% DMSO as a control. After treatment, the larvae were rinsed thoroughly, transferred to new water and left to develop until approximately stage 45, then euthanised by MS-222 overdose and fixed in modified Carnoy's solution as described above.

## Image capture and processing

Embryos/larvae were imaged using a Leica MZFLIII dissecting microscope fitted either with a QImaging MicroPublisher 5.0 RTV camera using QCapture Pro 7.0 software (QImaging) or a MicroPublisher 6 color CCD camera (Teledyne Photometrics) using Ocular software (Teledyne Photometrics). In most cases, focus stacking was performed using Helicon Focus software (Helicon Soft Limited) on image-stacks collected by manually focusing through the sample. Images were processed using Adobe Photoshop (Adobe Systems Inc).

## Statistical analysis

Initial data analysis was performed using Microsoft Excel. GraphPad Prism 10 (GraphPad Software, La Jolla, CA, USA) was used to compare datasets using a two-tailed Mann-Whitney (Wilcoxon rank sum) test and to generate scatter plots showing the median and interquartile range. The raw data are provided in *Supplementary files 2 and 3*.

## Acknowledgements

Thanks to Marek Rodina and Martin Kahanec for their help with sterlet spawns. This work was supported by the Anatomical Society and by the Biotechnology and Biological Sciences Research Council (BBSRC: grant BB/P001947/1 to CB). AC was supported by a PhD research studentship from the Anatomical Society with additional funding from the Cambridge Philosophical Society. Additional support for MM was provided by the Cambridge Isaac Newton Trust (grant 20.07[c] to CB) and by the School of the Biological Sciences, University of Cambridge. The work of RF, MV and MP was supported by the Ministry of Education, Youth and Sports of the Czech Republic projects CENAKVA (LM2018099) and Biodiversity (CZ.02.1.01/0.0/0.0/16_025/0007370) and by the Czech Science Foundation (22–31141 J).

## Additional information

### Funding

| Funder | Grant reference number | Author |
| --- | --- | --- |
| Anatomical Society | Research Studentship | Alexander S Campbell Clare VH Baker |
| Biotechnology and Biological Sciences Research Council | BB/P001947/1 | Clare VH Baker |
| Cambridge Philosophical Society | Research Studentship | Alexander S Campbell |
| Cambridge Isaac Newton Trust | Grant 20.07[c] | Clare VH Baker |
| Ministry of Education, Youth and Sports of the Czech Republic | projects CENAKVA [LM2018099] and Biodiversity [CZ.02.1.01/0.0/0.0/16_025/0007370] | Roman Franěk Michaela Vazačová Martin Pšenička |
| Czech Science Foundation | project 22-31141J | Roman Franěk Michaela Vazačová Martin Pšenička |
| School of the Biological Sciences, University of Cambridge | | Martin Minařík |

The funders had no role in study design, data collection and interpretation, or the decision to submit the work for publication.

### Author contributions

Alexander S Campbell, Formal analysis, Investigation, Visualization, Writing - original draft, Project administration, Writing – review and editing; Martin Minařík, Roman Franěk, Michaela Vazačová, Investigation, Writing – review and editing; Miloš Havelka, David Gela, Resources, Writing – review and editing; Martin Pšenička, Resources, Funding acquisition, Writing – review and editing; Clare VH Baker, Conceptualization, Supervision, Funding acquisition, Project administration, Writing – review and editing

### Author ORCIDs

Alexander S Campbell https://orcid.org/0009-0003-1539-214X

Martin Minařík ⓘ https://orcid.org/0000-0001-6660-0031
Roman Franěk ⓘ https://orcid.org/0000-0002-3464-1872
Clare VH Baker ⓘ https://orcid.org/0000-0002-4434-3107

### Ethics

Sterlet animal work was reviewed and approved by The Animal Research Committee of Research Institute of Fish Culture and Hydrobiology, Faculty of Fisheries and Protection of Waters, University of South Bohemia in České Budějovice, Vodňany, Czech Republic and Ministry of Agriculture of the Czech Republic (MSMT-12550/2016-3). Experimental fish were maintained according to the principles of the European Union (EU) Harmonized Animal Welfare Act of the Czech Republic, and Principles of Laboratory Animal Care and National Laws 246/1992 "Animal Welfare" on the protection of animals.

### Decision letter and Author response

Decision letter https://doi.org/10.7554/eLife.99798.sa1
Author response https://doi.org/10.7554/eLife.99798.sa2

## Additional files

### Supplementary files

Supplementary file 1. Breakdown by sgRNA mix of CRISPR/Cas9 experiments targeting *Bmp5* or *tyrosinase* (control). For each sgRNA mix, the table shows the number of independent batches, markers used for analysis, percentage with a phenotype, and genotyping information including genotyping primer sequences. *Note: Tyr* crispant data are shared between this study and *Minařík et al., 2024b*. *Tyr* sgRNAs 7 and 8 were designed and published by *Stundl et al., 2022* as their *tyr* sgRNA 3 and *tyr* sgRNA 4, respectively.

Supplementary file 2. Breakdown by individual crispant of CRISPR/Cas9 experiments targeting *Bmp5* or *tyrosinase* (control). For each crispant, the table lists the sgRNA mix used, the batch number, the ICE genotyping results and the number of ampullary organs within each field. Descriptive statistics are also shown for each ampullary organ field and across all ampullary organ fields for both *Bmp5* crispants and *Tyr* crispants.

Supplementary file 3. Breakdown of ampullary organ counting data for DMH1 experiments and older wild-type larvae. The table shows the number of ampullary organs within each field for each DMH1-treated embryo, DMSO-treated embryo (control) or older wild-type larva. Descriptive statistics are also given for each ampullary organ field and across all ampullary organ fields for DMH1-treated, DMSO-treated (control) and older wild-type larvae.

Supplementary file 4. Riboprobe information. For each riboprobe used during in situ hybridisation, the table lists the primer sequences used to clone the cDNA template, the GenBank accession number of the top-matched ohnolog, and the nucleotide region targeted by the riboprobe. It also shows the chromosomal location, the percentage identity of the riboprobe and the genome annotation of both the top-matched ohnolog and the second ohnolog (if present).

MDAR checklist

### Data availability

The original data required to reproduce the claims of the paper are provided in the manuscript and supplementary figures, together with additional image files deposited into the Dryad database (DOI: https://doi.org/10.5061/dryad.9s4mw6mt5). Previously published sterlet transcriptome assemblies (from pooled stage 40-45 sterlet heads; *Minařík et al., 2024a*) are available at DDBJ/EMBL/GenBank under the accessions GKLU00000000 (https://www.ncbi.nlm.nih.gov/nuccore/GKLU00000000) and GKEF01000000 (https://www.ncbi.nlm.nih.gov/nuccore/GKEF00000000.1). Previously published paddlefish RNA-seq data (from pooled paddlefish opercula and fin tissue at stage 46; *Modrell et al., 2017a*) are available via the NCBI Gene Expression Omnibus (GEO) database (https://www.ncbi.nlm.nih.gov/geo/) under accession code GSE92470.

The following dataset was generated:

| Author(s) | Year | Dataset title | Dataset URL | Database and Identifier |
|---|---|---|---|---|
| Campbell AS, Minařík M, Franěk R, Vazačová M, Havelka M, Gela D, Pšenička M, Baker CVH | 2025 | Data for: Opposing roles for Bmp signalling during the development of electrosensory lateral line organs. | https://doi.org/10.5061/dryad.9s4mw6mt5 | Dryad, 10.5061/dryad.9s4mw6mt5 |

The following previously published datasets were used:

| Author(s) | Year | Dataset title | Dataset URL | Database and Identifier |
|---|---|---|---|---|
| Minařík M, Modrell MS, Gillis JA, Campbell AS, Fuller I, Lyne R, Micklem G, Gela D, Pšenička M, Baker CVH | 2023 | TSA: Acipenser ruthenus, transcriptome shotgun assembly | https://www.ncbi.nlm.nih.gov/nuccore/GKLU00000000.1 | NCBI GenBank, GKLU00000000.1 |
| Minařík M, Modrell MS, Gillis JA, Campbell AS, Fuller I, Lyne R, Micklem G, Gela D, Pšenička M, Baker CVH | 2023 | TSA: Acipenser ruthenus, transcriptome shotgun assembly | https://www.ncbi.nlm.nih.gov/nuccore/GKEF00000000.1 | NCBI GenBank, GKEF00000000.1 |
| Modrell MS, Lyne M, Carr AR, Zakon HH, Buckley D, Campbell AS, Davis MC, Micklem G, Baker CVH | 2017 | Insights into electrosensory organ development, physiology and evolution from a lateral line organ-enriched transcriptome | https://www.ncbi.nlm.nih.gov/geo/query/acc.cgi?acc=GSE92470 | NCBI Gene Expression Omnibus, GSE92470 |

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

# Appendix 1

## Appendix 1—key resources table

| Reagent type (species) or resource | Designation | Source or reference | Identifiers | Additional information |
|---|---|---|---|---|
| Gene (*Acipenser ruthenus*) | *Acvr2a* | NCBI_Gene (RRID:SCR_002473) | GeneID: 117427134 | |
| Gene (*A. ruthenus*) | *Apcdd1* | NCBI_Gene (RRID:SCR_002473) | GeneID: 117394164 | |
| Gene (*A. ruthenus*) | *Bmp4* | NCBI_Gene (RRID:SCR_002473) | GeneID: 117395489 | |
| Gene (*A. ruthenus*) | *Bmp5* | NCBI_Gene (RRID:SCR_002473) | GeneID: 117402601 | |
| Gene (*A. ruthenus*) | *Cacna1d* | NCBI_Gene (RRID:SCR_002473) | GeneID:117413950 | |
| Gene (*A. ruthenus*) | *Kcnab3* | NCBI_Gene (RRID:SCR_002473) | GeneID:117404443 | |
| Gene (*A. ruthenus*) | *Sostdc1* | NCBI_Gene (RRID:SCR_002473) | GeneID:117400407 | |
| Biological sample (*A. ruthenus*) | Fertilised sterlet sturgeon eggs and embryos/larvae (*A. ruthenus*) | Research Institute of Fish Culture and Hydrobiology, Faculty of Fisheries and Protection of Waters, University of South Bohemia in České Budějovice Vodňany, Czech Republic | | |
| Antibody | anti-Sox2 (rabbit monoclonal) | Abcam | Cat.#:ab92494; RRID:AB_10585428 | (1:200) |
| Antibody | anti-human Phospho-SMAD1/5/9 (rabbit monoclonal) | Cell Signalling Technology | Cat.#:13820; RRID:AB_2493181 | (1:100) |
| Antibody | horseradish peroxidase-conjugated goat anti-rabbit IgG | Jackson ImmunoResearch | Cat.#:111-035-003; RRID:AB_2313567 | (1:300) |
| Recombinant DNA reagent | pX335-U6-Chimeric_BB-CBh-hSpCas9n(D10A) (plasmid) | Addgene (**Cong et al., 2013**; DOI: 10.1126/science.1231143) | RRID:Addgene_42335 | Used to synthesize DNA templates containing the single guide (sg)RNA scaffold |
| Sequence-based reagent | *Bmp4* riboprobe forward primer (F) | This paper | PCR primers | GGGGCCGCAAGAAAAACCGGAA |
| Sequence-based reagent | *Bmp4* riboprobe reverse primer (R) | This paper | PCR primers | TCGCAGTGAGCCTTGCCCATTT |
| Sequence-based reagent | *Bmp5* riboprobe F | This paper | PCR primers | ACCCAGTGGTTGCTTGTAGC |
| Sequence-based reagent | *Bmp5* riboprobe R | This paper | PCR primers | ATTCTGGGCTTACCACATCG |
| Sequence-based reagent | *Cacna1d* riboprobe F | This paper | PCR primers | GCCACGAACATCACTCCGCCAA |
| Sequence-based reagent | *Cacna1d* riboprobe R | This paper | PCR primers | TCATGTCGCAAGCGTCCGCAAT |
| Sequence-based reagent | *Kcnab3* riboprobe F | **Minařík et al., 2024a** (DOI: https://doi.org/10.3389/fcell.2024.1327924) | PCR primers | GGTAAATTCAGCGTGGAGGA |
| Sequence-based reagent | *Kcnab3* riboprobe R | **Minařík et al., 2024a** (DOI: https://doi.org/10.3389/fcell.2024.1327924) | PCR primers | ACCTTCGATGATGTGCTTCC |
| Sequence-based reagent | *Sostdc1* riboprobe F | This paper | PCR primers | CCACGCCTGGTTAATCCTGTGGA |
| Sequence-based reagent | *Sostdc1* riboprobe R | This paper | PCR primers | GTGCTTGCCCGTCTTGCCTGAT |
| Sequence-based reagent | sgRNA scaffold R | Pers. comm., Dr Ahmed Elewa, Karolinska Institutet, Stockholm, Sweden | PCR primers | AAAAAAGCACCGACTCGGTGCC |
| Sequence-based reagent | *Bmp5* sgRNA F1 | This paper | PCR primers | GATCACTAATACGACTCACTATAGTCACGCAGAAAAGCACAGGGGTTTAGAGCTAGAAAT |
| Sequence-based reagent | *Bmp5* sgRNA F2 | This paper | PCR primers | GATCACTAATACGACTCACTATAGAGATGATGCCTGTTTGCCAGGTTTTAGAGCTAGAAAT |
| Sequence-based reagent | *Bmp5* sgRNA F3 | This paper | PCR primers | GATCACTAATACGACTCACTATAGGCAAACGAGGAGGAAAACGGTTTTAGAGCTAGAAAT |
| Sequence-based reagent | *Bmp5* sgRNA F4 | This paper | PCR primers | GATCACTAATACGACTCACTATAGTACAATGCCATGGCAAACGGTTTTAGAGCTAGAAAT |

*Appendix 1 Continued on next page*

*Appendix 1 Continued*

| Reagent type (species) or resource | Designation | Source or reference | Identifiers | Additional information |
|---|---|---|---|---|
| Sequence-based reagent | *Tyr* sgRNA F1 | **Minařík et al., 2024b** (DOI: https://doi.org/10.1101/2023.04.15.537030) | PCR primers | GATCACTAATACGACTCACTAT AGGTGCCAAGGCAAAAACGC TGTTTTAGAGCTAGAAAT |
| Sequence-based reagent | *Tyr* sgRNA F2 | **Minařík et al., 2024b** (DOI: https://doi.org/10.1101/2023.04.15.537030) | PCR primers | GATCACTAATACGACTCACTAT AGATATCCCTCCATACATTATG TTTTAGAGCTAGAAAT |
| Sequence-based reagent | *Tyr* sgRNA F3 | **Minařík et al., 2024b** (DOI: https://doi.org/10.1101/2023.04.15.537030) | PCR primers | GATCACTAATACGACTCACTAT AGATGTTTCTAAACATTGGGG GTTTTAGAGCTAGAAAT |
| Sequence-based reagent | *Tyr* sgRNA F4 | **Minařík et al., 2024b** (DOI: https://doi.org/10.1101/2023.04.15.537030) | PCR primers | GATCACTAATACGACTCACTA TAGCTATGAATTTATTTTTTC GTTTTAGAGCTAGAAAT |
| Sequence-based reagent | *Tyr* sgRNA F5 | **Minařík et al., 2024b** (DOI: https://doi.org/10.1101/2023.04.15.537030) | PCR primers | GATCACTAATACGACTCACTAT AGCAAGGTATACGAAAGTTGA GTTTTAGAGCTAGAAAT |
| Sequence-based reagent | *Tyr* sgRNA F6 | **Minařík et al., 2024b** (DOI: https://doi.org/10.1101/2023.04.15.537030) | PCR primers | GATCACTAATACGACTCACTAT AGATTGCAAGTTCGGCTTCTT GTTTTAGAGCTAGAAAT |
| Sequence-based reagent | *Tyr* sgRNA F7 | **Stundl et al., 2022** (DOI: https://doi.org/10.3389/fcell.2022.750833) | PCR primers | GATCACTAATACGACTCACTAT AGGTTAGAGACTTTATGTAAC GTTTTAGAGCTAGAAAT |
| Sequence-based reagent | *Tyr* sgRNA F8 | **Stundl et al., 2022** (DOI: https://doi.org/10.3389/fcell.2022.750833) | PCR primers | GATCACTAATACGACTCACTAT AGGCTCCATGTCTCAAGTCC AGTTTTAGAGCTAGAAAT |
| Sequence-based reagent | *Bmp5* genotyping F | This paper | PCR primers | GGAACACAGTCGCTGAAGTG |
| Sequence-based reagent | *Bmp5* genotyping R | This paper | PCR primers | GTTGCATACATGCCCAGATG |
| Sequence-based reagent | *Tyr* genotyping F1 | **Minařík et al., 2024b** (DOI: https://doi.org/10.1101/2023.04.15.537030) | PCR primers | GCGTCTCTCCAGTCCCAATA |
| Sequence-based reagent | *Tyr* genotyping R1 | **Minařík et al., 2024b** (DOI: https://doi.org/10.1101/2023.04.15.537030) | PCR primers | AGAGAGAAGTGGCCCTTGGT |
| Peptide, recombinant protein | Cas9 protein with NLS | PNA Bio | Cat.#:CP01-200 | |
| Peptide, recombinant protein | Q5 High-Fidelity DNA Polymerase | New England Biolabs | Cat.#:M0491S | |
| Commercial assay or kit | Ambion Turbo DNA-free kit | Thermo Fisher Scientific | Cat.#:AM1907 | |
| Commercial assay or kit | High-Capacity cDNA Reverse Transcription Kit | Applied Biosystems | Cat.#:4368814 | |
| Commercial assay or kit | QIAGEN PCR Cloning Kit | QIAGEN | Cat.#:231124 | |
| Commercial assay or kit | MinElute Gel Extraction Kit | QIAGEN | Cat.#:28604 | |
| Commercial assay or kit | EnzMet kit | Nanoprobes | Cat.#:6010 | |
| Commercial assay or kit | PCRBIO Rapid Extract PCR Kit | PCR Biosystems | Cat.#:PB10.24–08 | |
| Commercial assay or kit | HS Taq Mix Red | PCR Biosystems | Cat.#:PB10.23–02 | |
| Commercial assay or kit | Monarch PCR & DNA Cleanup Kit | New England Biolabs | Cat.#:T1030 | |
| Commercial assay or kit | Monarch RNA Cleanup Kit | New England Biolabs | Cat.#:T2040 | |
| Commercial assay or kit | HiScribe T7 High Yield RNA Synthesis Kit | New England Biolabs | Cat.#:E2040S | |
| Chemical compound, drug | DMH1 (dorsomorphin homolog 1) | Cayman Chemical | Cat.#:CAY16679 | |
| Software, algorithm | Benchling | Benchling | RID:SCR_013955 | |
| Software, algorithm | Primer3Plus | Primer3Plus | RRID:SCR_003081 | |
| Software, algorithm | Inference of CRISPR Edits (ICE) | Synthego | RRID:SCR_024508 | |
| Software, algorithm | NCBI BLAST | National Institutes of Health (NIH) | RRID:SCR_004870 | |

*Appendix 1 Continued on next page*

*Appendix 1 Continued*

| Reagent type (species) or resource | Designation | Source or reference | Identifiers | Additional information |
|---|---|---|---|---|
| Software, algorithm | Helicon Focus | Helicon Soft | RRID:SCR_014462 | |
| Software, algorithm | Adobe Photoshop | Adobe | RRID:SCR_014199 | |
| Software, algorithm | GraphPad Prism | GraphPad | RRID:SCR_002798 | |
| Software, algorithm | QCapture Pro 7.0 | QImaging | RRID:SCR_014432 | |
| Software, algorithm | Ocular | Teledyne Photometrics | RRID:SCR_024490 | |

