## [Editor Report]

This fundamental study provides new insight into the molecular signalling pathways that govern the formation of electrosensory ampullary organs in a non-model organism, the sterlet sturgeon. By using a combination of targeted gene knock-out and chemical inhibition, the study provides overall convincing evidence for differential roles of BMP signaling in lateral-line development. The study is particularly helpful for understanding the development of a still enigmatic sensory system, and for its evolutionary implications.

---

## [Decision Letter]

**Decision letter after peer review:**

Thank you for submitting your article "Two opposing roles for Bmp signalling in the development of electrosensory lateral line organs" for consideration by *eLife*. Your article has been reviewed by 2 peer reviewers, and the evaluation has been overseen by a Reviewing Editor and Kathryn Cheah as the Senior Editor.

There are specific suggestions by both reviewers that we would ask you to address, either experimentally or by discussion.

1. It remains unclear if the opposing effects of Bmp can be adequately accounted for by differences in duration of Bmp inhibition or distinct functions of different Bmp ligand expressed at different times. The authors should check inhibition of bmp with DMH1 at a later point recapitulates the effects seen in Bmp5 crispants.

2. It is not clear how and if the manipulations affect P-Smad expression following the two manipulations- do they both reduce P-Smad expression or do they have opposing effects?

3. Do changes in the expression of sostdc1 or apcdd1 in the two different manipulations to reduce BMP provide any additional information that might account for differential effects seen in Bmp5 crispants and following DMH1 treatment.

---

## [Author Response]

Essential revisions:There are specific suggestions by both reviewers that we would ask you to address, either experimentally or by discussion.1. It remains unclear if the opposing effects of Bmp can be adequately accounted for by differences in duration of Bmp inhibition or distinct functions of different Bmp ligand expressed at different times. The authors should check inhibition of bmp with DMH1 at a later point recapitulates the effects seen in Bmp5 crispants.

This is an interesting suggestion but we are unable to do further experiments at this time. We have amended the wording in the first paragraph of the Discussion section "Bmp5 promotes ampullary organ formation in sterlet" to emphasise that we do not know when Bmp5 acts and to suggest the proposed experiment:

Lines 850-4: "Its precise function and timing of action in promoting ampullary organ formation remain to be determined. Although beyond the scope of this study, it could be informative to test whether inhibiting Bmp signalling using DMH1 for discrete periods at successively later time-points recapitulates the *Bmp5* crispant phenotype."

2. It is not clear how and if the manipulations affect P-Smad expression following the two manipulations- do they both reduce P-Smad expression or do they have opposing effects?

A priori, both manipulations will reduce Bmp signalling pathway activity, hence will reduce phospho-SMAD1/5/9 expression. DMH1 blocks signalling through the type I Bmp receptors Acvr1 (Alk2), Acvrl1 (Alk1) and Bmpr1a, all of which signal via Smad1/5/9, hence DMH1 treatment will directly reduce phospho-SMAD1/5/9 expression. A priori, CRISPR/Cas9-mediated targeting of *Bmp5*, leading to reduced expression of a Bmp ligand, will reduce Bmp signalling pathway activity in the responding cells (relative to controls), hence will reduce phospho-SMAD1/5/9 expression in the responding cells.

In principle, the reduced Bmp signalling that is the immediate outcome of both manipulations could indirectly lead to increased Bmp signalling (hence increased phospho-SMAD1/5/9 expression) at a later time-point (relative to controls) for example, via reduced expression/activity of a Bmp inhibitor. In practice, especially given the mosaicism of CRISPR/Cas9 phenotypes, we think it would be difficult to test for the potential existence of such indirect, longer-term opposing effects via changes in phospho-SMAD1/5/9 expression.

3. Do changes in the expression of sostdc1 or apcdd1 in the two different manipulations to reduce BMP provide any additional information that might account for differential effects seen in Bmp5 crispants and following DMH1 treatment.

This is another interesting idea, but we are unable to do further experiments at this time. We also note that crispant mosaicism would make this hard to test in practice. We have added this suggestion to both Discussion sections, as follows:

Discussion section "Bmp5 promotes ampullary organ formation in sterlet", lines 854-6:

"Similarly, it would be interesting to see whether developing ampullary organs in *Bmp5* crispants show any changes in the expression of the Bmp/Wnt inhibitor genes *Sostdc1* and/or *Apcdd1*."

Discussion section "Bmp5 promotes ampullary organ formation in sterlet", lines 965-7:

"However, *Bmp5* is expressed in mature ampullary organs and neuromasts and additional as-yet unidentified Bmp ligand genes may also be expressed, given the more extensive pattern of pSMAD1/5/9 immunoreactivity. Furthermore, the dual Bmp/Wnt inhibitor genes *Sostdc1* and *Apcdd1* are both expressed during ampullary organ development: their roles are unknown, and the expression of one or both genes could be regulated by Bmp signalling. Overall, the precise mechanism by which Bmp signalling normally prevents supernumerary and ectopic ampullary organ formation remains to be established…"